# Kinesin-1 activity recorded in living cells with a precipitating dye

Simona Angerani[1], Eric Lindberg[1], Nikolai Klena [2], Christopher K. E. Bleck [3], Charlotte Aumeier[1✉] & Nicolas Winssinger [1✉]

Kinesin-1 is a processive motor protein that uses ATP-derived energy to transport a variety of intracellular cargoes toward the cell periphery. The ability to visualize and monitor kinesin transport in live cells is critical to study the myriad of functions associated with cargo trafficking. Herein we report the discovery of a fluorogenic small molecule substrate (QPD-OTf) for kinesin-1 that yields a precipitating dye along its walking path on microtubules (MTs). QPD-OTf enables to monitor native kinesin-1 transport activity in cellulo without external modifications. In vitro assays show that kinesin-1 and MTs are sufficient to yield fluorescent crystals; in cells, kinesin-1 specific transport of cargo from the Golgi appears as trails of fluorescence over time. These findings are further supported by docking studies, which suggest the binding of the activity-based substrate in the nucleotide binding site of kinesin-1.

[1] School of Chemistry and Biochemistry, NCCR Chemical Biology, Faculty of Science, University of Geneva, Geneva, Switzerland. [2] Department of Cell Biology, Faculty of Science, University of Geneva, Geneva, Switzerland. [3] Electron Microscopy Core Facility, National Heart, Lung and Blood Institute, National Institutes of Health, Bethesda, MD, USA. ✉email: charlotte.aumeier@unige.ch; nicolas.winssinger@unige.ch

Microtubules (MTs) are polymers of α and β tubulin that are involved in several functions in cells. Although the majority of MTs emanates from the centrosome[1], the main non-centrosomal MT organizing center is represented by the Golgi apparatus[2]. The minus-end of the MT is anchored at the MT organizing center and the dynamic plus-end orientated towards the cell periphery.

Motor proteins, such as kinesins and dyneins, are ATPases that bind to MTs and walk along with them in response to cargo binding[3]. Kinesin-1 is a member of the kinesin family that transports cargoes to the cell periphery walking on MTs towards their plus-end[4]. Among others, kinesin-1 is interacting with both pre-Golgi and Golgi membranes and it is involved in Golgi-to-ER and ER-to-Golgi trafficking[5,6]. Kinesin-1 is autoinhibited and only functionally active once bound to cargo during Golgi-to-ER transport. Active kinesin-1 molecules bound to microtubules run across hundreds of tubulin dimers without dissociating[7,8]. Truncation of the kinesin-1 heavy chain can lead to constitutively active mutants of kinesin-1[9,10]. The motion of kinesin-1 occurs preferentially on a subset of modified, long-lived MTs, such as acetylated and detyrosinated MTs[11–13]. The transport activity of kinesin-1 can be inhibited by Taxol, a drug that stabilizes and changes the MT structure[14].

To date, techniques to monitor motor proteins in cells have relied on antibodies, quantum dots, or on engineered versions of the motors bearing fluorescent tags[15–20]; these techniques require sample treatment (fixation and staining) or manipulation (transfection). Moreover, these techniques stain total protein content, irrespectively of their motility. Only about 30 % of kinesin-1 is active in cells[21]; this makes it difficult to study kinesin-1-GFP movement along MTs within the strong background of immotile kinesin-1-GFP in transfected cells[11].

QPD is a quinazolinone-based precipitating dye developed to easily visualize enzymatic activity in cellulo[22,23]. Accordingly, QPD has been used to design fluorogenic reporters of phosphatase (PO$_4^-$ derivative)[24], protease (ester derivative)[25], and H$_2$O$_2$ (boronic acid derivate)[26] or catalysis (azide[27] or picolinium[28] derivative), and these substrates have been used to label a number of organelles and cytoskeletal elements[24,27]. QPD fluorescence derives from an excited-state intramolecular proton transfer (ESIPT)[29] between the phenolic group and the quinazolinone.

Functionalization of the phenol frees the aryl moiety out of planarity with the quinazolinone, which dramatically reduces its aggregation and precipitation; derivatization with a polar group renders these molecules water soluble.

Herein, we report the discovery of a QPD derivative (QPD-OTf) that acts as an activity-based fluorogenic substrate for kinesin-1 by producing a precipitating fluorescent dye along its walking path on MTs. The phenolic moiety is functionalized with a triflate group that renders the molecule soluble in an aqueous buffer and non-fluorescent. Biochemical experiments show that kinesin-1 and MTs are sufficient to yield fluorescent crystals and that inhibition of kinesin-1's ATPase activity reduces the formation of fluorescent crystals. Docking studies support the binding of the activity-based substrate in the nucleotide-binding site, aligning the triflate leaving group with the gamma-phosphate group of ATP. In live cells, the crystals are centered in the Golgi apparatus and radially elongate towards the cell periphery, recording kinesin-1 motion on MTs. Thus, QPD-OTf enables visualization of the native transport activity of kinesin-1 in cellulo without external modifications.

## Results

**QPD-OTf forms crystals in living cells**. Taking advantage from the large applicability of QPD-based profluorophores, we envisioned the synthesis of a QPD derivative, QPD-OTf (Fig. 1a) initially designed to be responsive to superoxide, for the visualization of oxidative stress in cellulo. In analogy with a reported fluorescein derivative[30], the trifluoromethanesulfonate ester should be activated enough to undergo a nucleophilic attack by O$_2^{\cdot-}$ affording the free phenol. Surprisingly, QPD-OTf was found not to be responsive to O$_2^{\cdot-}$ in vitro, with no precipitation observed.

QPD-OTf treatment of zymosan stimulated RAW264.7 cells caused dotted fluorescent precipitate after 10 min that evolved into complex filamentous crystals within 1 h (Supplementary Fig. 1). The QPD crystal is an extended, aster-like fluorescent crystals expanding throughout the cell (Supplementary Fig. 2 and Supplementary Movie 1) and even able to deform the cell membrane (Fig. 1b and Supplementary Fig. 3). While overnight exposure of 10–20 µM QPD-OTf induces cell death, temporary

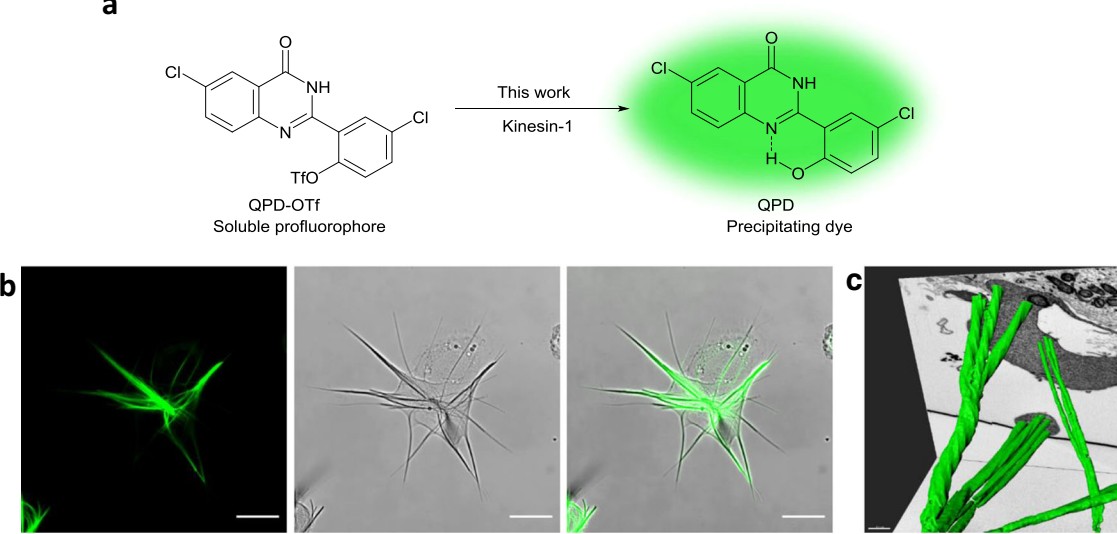

**Fig. 1 Schema of the soluble profluorophore QPD-OTf and the insoluble fluorescent dye QPD and QPD crystal in cells. a** Structure of QPD-OTf and QPD. **b** Aster-like QPD crystal in live U2OS cell (20 µM QPD-OTf, 4 h); green: crystal. Left: QPD fluorescence; middle: bright field image; right: merged channels. Scale bar 20 µm. **c** FIB-SEM 3D-reconstruction of the crystal inside HeLa cells; green: crystal. Scale bar 0.4 µM.

exposure of up to 4 h followed by fresh media replacement, preserves cell viability almost completely (Supplementary Fig. 4) and even dissolved the crystal over time (Supplementary Movie 2 and Supplementary Fig. 5). Since the fluorescent signal can only arise from a QPD displaying an uncaged phenol, we hypothesize that the triflate caging moiety must be removed inside the cell upon enzymatic activity. The observation of these crystals across multiple cell lines from different mammalian species (RAW264.7, HeLa, MCF-7, HEK293, U2OS, PTK2) shows that this activity is conserved and not restricted to a specialized cell line (Supplementary Fig. 3). While different cell lines afforded slightly different crystal morphology, all cell lines showed fibers that emanate from central points. It should be noted that different cell lines also showed different kinetics of crystal formation which can in part account for the differences in crystals when comparing the same time point across different cell lines.

FIB-SEM analysis of HeLa cells treated with QPD-OTf (20 μM, 4 h) showed that the crystals have a well-defined organization, with rotational symmetry order 3-like structure (Fig. 1c), and hexagonal cross-section, whose size varies from 100 to 700 nm (Supplementary Fig. 6). It should be noted that the rigidity of crystals is such that plasma membranes of retracting cells are deformed (Fig. 1c). In the FIB-SEM image, a more extreme case is observed where the crystal penetrates through the nucleus (Fig. 1c and Supplementary Fig 6a). Given the incubation time, fixation, and dehydration steps involved in the sample preparation, this observation may be an artifact of sample preparation. We also noted that the crystal has a clear nucleation center (Supplementary Fig. 6) which spurred us to further investigate the triggering mechanism behind the crystal formation.

**Crystals co-localize with MTs.** Many crystals are localized at the cell center, spanning with their filamentous nature throughout the cell. Due to their organization and architecture, we thought that the enzymatic activity generating QPD-crystals might be linked to the actin or MT cytoskeleton. Labeling the cytoskeleton after QPD-OTf treatment showed clear colocalization between the crystal and the MT network (Fig. 2) and only marginal correlation with actin (Supplementary Fig. 7). In fixed cells, immunostaining of α-tubulin showed alignment of crystal fibers along with MT bundles (Fig. 2a). We observed that only a distinct subset of the MT network seemed to co-localize with the crystal. Live-cell imaging by expressing GFP-tubulin in Ptk2 (Fig. 2b, e) and HeLa cells (Fig. 2c, d, f and Supplementary Fig. 8), or staining MTs with SiR-Tubulin (Supplementary Fig. 9)[31], a Taxol-based fluorescent dye, confirmed the colocalization of the crystal with a subset of the MT network.

Not all MTs within the cellular network have the same dynamical properties, and we wondered if MT dynamics was linked to crystal formation upon QPD-OTf treatment. In order to test our hypothesis, we altered MT dynamics and studied the impact on crystal formation. First, we treated U2OS cells with 1 μM Taxol and 20 μM QPD-OTf for 4 h and compared to untreated cells (control). As shown in Fig. 3a and quantified in Fig. 3b, Taxol-induced MT stabilization reduced the crystal formation by 75%. Moreover, the few crystals we observed in the Taxol-treated sample were much thinner than in the control (Fig. 3a, zoom). Second, we completely depolymerized the MT network by cold treatment, followed by 20 μM QPD-OTf incubation for 4 h. In this case, no crystals were observed (Fig. 3a, middle). The almost complete absence of crystals with both treatments suggests that the integrity and physiological dynamic of MTs are substantial requirements for crystal development.

**QPD crystals form along MTs originating from the Golgi apparatus.** To identify the origin of the specific localization of the

crystal within a subset of the MT network, we focused on the nucleation site of the crystals. In fact, most of the cells in interphase show a single crystal, originating close to the nucleus. This raises the possibility that the nucleation site of the crystal overlaps with the nucleation site of MTs. MTs nucleate mainly from MT organizing centers located close to the cell center. However, the most prominent MT organizing center, the centrosome, did not co-localize with the triggering site for QPD precipitation (Supplementary Fig. 10). Therefore, we investigated another MT organizing center: the Golgi apparatus. The Golgi is known to be involved in MT nucleation[2,32], and to be a key player in the secretory pathway[33].

To assess whether the MT organizing center at the Golgi triggers QPD precipitation, we visualized the Golgi in U2OS cells by transfecting them with mCherry-Giantin. Confocal fluorescence microscopy revealed that the crystals are nucleated at the Golgi apparatus (Fig. 4a and Supplementary Movie 3) and that Golgi vesicles are found along the crystal filaments (Supplementary Fig. 11). The observation was confirmed by transfecting mCherry-Giantin, a Golgi-marker, in a stable expressing PTK2-GFP-tubulin cell line with subsequent treatment of QPD-OTf (Supplementary Fig. 12). The MT organizing center of the Golgi, together with its transport activity could therefore play a key role in determining the selective transformation of QPD-OTf to QPD in a specific cellular location.

In order to further investigate the role of the Golgi apparatus in the formation of QPD crystals, we studied the effect of Brefeldin A (BFA), an inhibitor of Golgi trafficking. BFA impairs the function of Golgi, resulting in its fragmentation[34,35]. While the Golgi apparatus appears as a compact complex in U2OS interphase cells, treatment with 20 μM BFA showed the expected scattered Golgi fragments (Supplementary Fig. 13). QPD-OTf addition to BFA treated cells resulted in thinner crystals (fiber thickness reduced by 58%) (Fig. 4b, c) with multiple foci of origin instead of only one as in control cells (Fig. 4b). Although the Golgi was fragmented, the nucleation site of the crystal remained co-localizing with the Golgi (Fig. 4d). This shows that the Golgi apparatus is intimately linked to crystal formation and that modifications of the Golgi structure correlate with crystal morphology and location.

**Purified MTs are not sufficient to generate crystals in vitro.** Having established that MT dynamics is necessary for the development of crystals, and that the Golgi apparatus, known as an MT organizing center, dictates the location of the crystals, we assessed whether pure MT polymerization is sufficient to generate a crystal in vitro. To this end, we tried to precipitate QPD on Taxol stabilized MTs, or on dynamic MTs elongating from stabilized seeds. No crystal formation could be observed and no fluorescence of QPD was detected in our in vitro TIRF assay, even after 2 h (Supplementary Fig. 14). Thus, we reasoned that the conversion of QPD-OTf to QPD crystals must be triggered by an enzymatic event that is closely related to and dependent on the MT network, but an activity that is not essential for MT elongation. With these considerations, we directed our attention to motor proteins since these proteins move cargoes along MTs in an energy-dependent manner.

**QPD-OTf conversion to QPD depends on kinesin-1 motility.** The plus-end-directed motor protein kinesin-1 transports cargo from the Golgi to the ER and its enzymatic activity might be responsible for the conversion of QPD-OTf to QPD with ensuing crystal formation. We therefore genetically modified kinesin-1 activity in cells and analyzed the effect of kinesin-1 activity on crystal formation. Cells were treated with QPD-OTf after

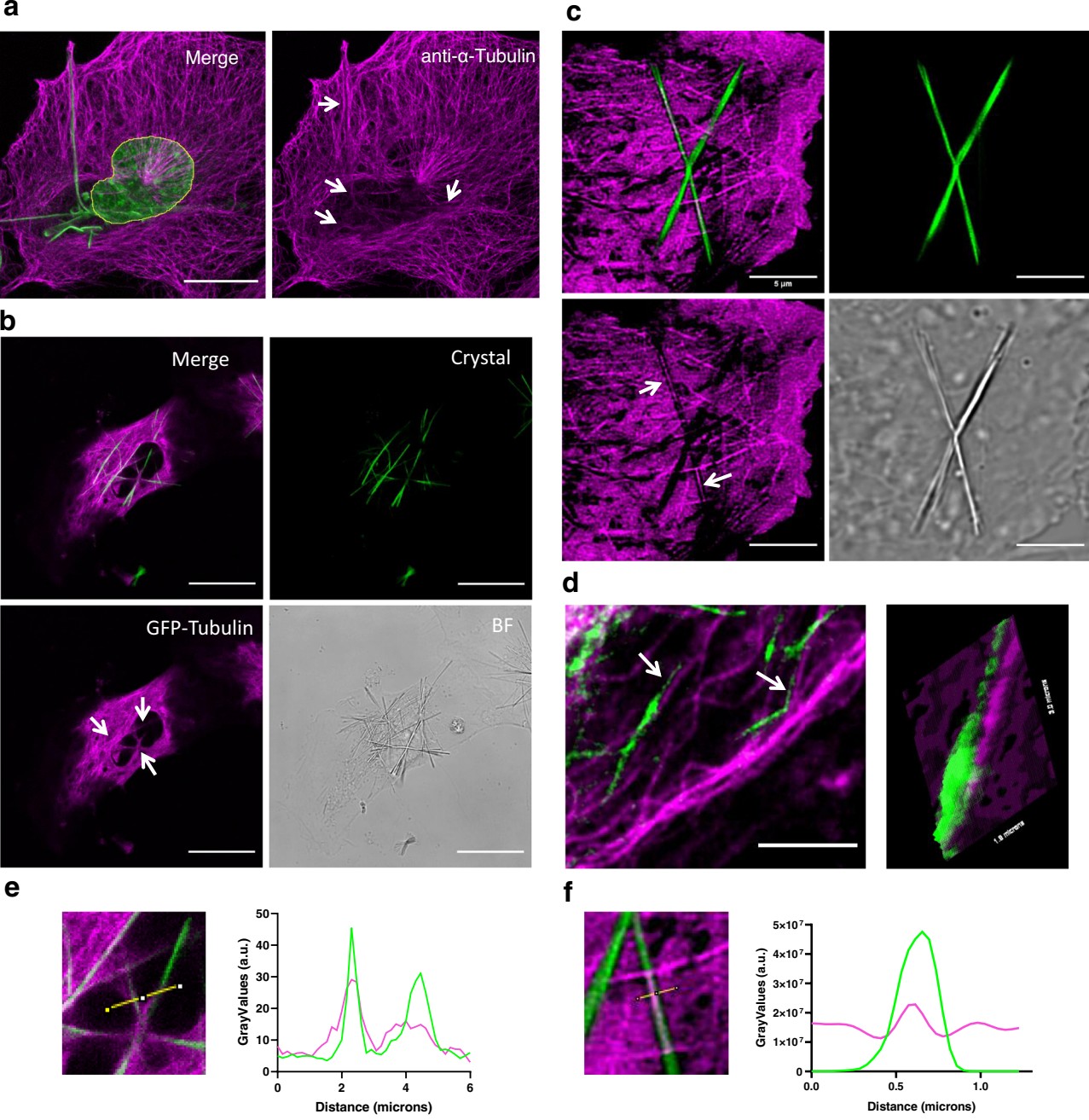

**Fig. 2 QPD-OTf forms crystals that strongly co-localize with MTs in cells. a** Tubulin immunostaining in fixed U2OS treated with QPD-OTf (20 µM; 4 h); white arrows indicate colocalization with crystals (anti-α-tubulin: magenta; crystal and DAPI: green; the nucleus is contoured in yellow). Scale bar 20 µM. **b** Live-cell imaging of PTK2-GFP-Tubulin treated with QPD-OTf (20 µM; 2 h); white arrows indicate colocalization with crystals (GFP-tubulin: magenta; crystal: green). Scale bar 20 µM. **c** Super-resolution imaging of HeLa-GFP-Tubulin live cells treated with QPD-OTf (20 µM; 2 h); white arrows indicate colocalization with crystals (GFP-tubulin: magenta; crystal: green). Scale bar 5 µm. **d** Super-resolution image of live HeLa-GTP-Tubulin cells treated with QPD-OTf at an early time point (QPD-OTf 20 µM; 20 min) (left); white arrows indicate colocalization with crystals; scale bar 5 µm. Surface plot of crystal and MT signal (right) (GFP-tubulin: magenta; crystal: green); surface section: 1.8 × 3.0 microns. **e**, **f** Plot profiles of tubulin channel (magenta) and QPD channel (green). Yellow lines represent the sections plotted in the graphs; (a.u. represent arbitrary units). Source data are provided as Source Data file.

transfection with Kin330-GFP or Kin560-GFP, two truncated versions of kinesin-1 fused to GFP displaying no ability to walk on MTs, and constitutively active walking on MTs respectively[36,37]. Cells transfected with the kinesin-1 mutant Kin330 showed a reduction in crystal numbers by 87% compared to non-transfected cells (Fig. 5a, c and Supplementary Fig. 15), consistent with the inhibitory effect of Kin330 on the functional activity of native kinesin-1[38]. The residual formation of some crystals could be attributed to the activity of endogenous wildtype

kinesin-1. Although over-activity of Kin560 in cells transfected with Kin560-GFP did not further increase crystal formation (Fig. 5b, c and Supplementary Fig. 15), it was possible to correlate the crystal filaments to the signal of Kin560-GFP on MTs (Fig. 5b, zoom). It is also noteworthy that despite the concentration of Kin560-GFP and broad distribution, fluorescent crystals are only seen on specific tubulin axis. In order to validate that the crystal formation correlates with the activity of kinesin-1, we performed a siRNA knockdown of kinesin-1. We observed that the intensity

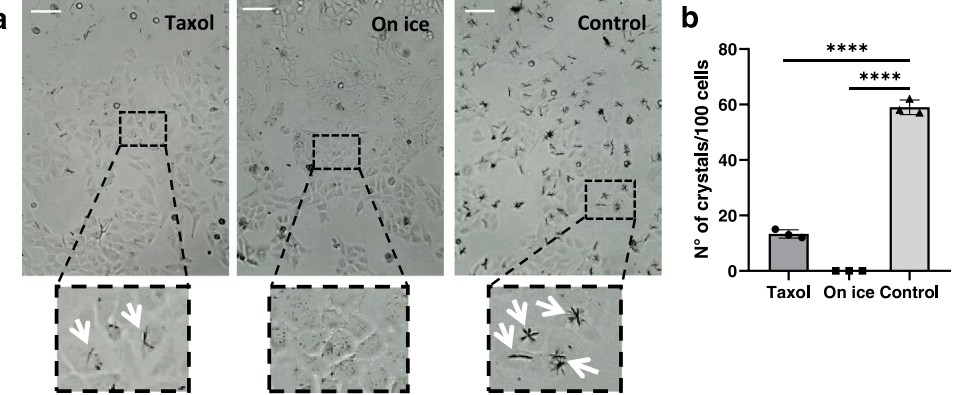

**Fig. 3 Formation of QPD crystals in U2OS live cells is disrupted by induced microtubule stabilization or depolymerization. a** Representative images of crystal formation in cells treated (left) with 1 µM Taxol for 1 h and 20 µM QPD-OTf for 4 h at 37 °C; (middle) on ice for 1 h and with 20 µM QPD-OTf for 4 h on ice; and (right) with 20 µM QPD-OTf for 4 h at 37 °C. (Bottom) Zoomed-in images of cells in the black squares; white arrows indicate crystals. Scale bar 100 µM. **b** Quantification of the number of crystals for conditions reported in **a**; *n* = 100; data are presented as mean value ± the standard deviation (SD); data are the average of three independent experiments; Statistics were calculated using a two-tailed *t*-test; ****$p$ < 0.0001. Source data are provided as Source Data file.

of the crystals was significantly reduced in the siRNA treated sample compared to the control (Fig. 5d, e). The knockdown efficiency was confirmed by western blot (Fig. 5f).

To further investigate the effect of kinesin-1 activity on crystal formation, we tested the effect of adding kinesore, a small molecule kinesin-1 activator[39]. In cells, kinesin-1 is inactive and only gets activated upon cargo binding[40,41]. Kinesore interacts with kinesin-1 at the kinesin light chain-cargo interface ($K_i$ = 49 µM for aiKLC2$^{TPR}$: SKIP$^{WD}$ complex), mimicking the effect of cargo binding and resulting in kinesin-1 activation. The enhanced motion causes profound rearrangement of the MT network. We found that the addition of 100 µM kinesore to U2OS cells, followed by incubation with 20 µM QPD-OTf inhibited the formation of crystals, yet generated some diffuse QPD fluorescence (Fig. 5g, h). This diffused fluorescence as a result of kinesore treatment is attributed to the over-activity of kinesin-1, with a motor activity that is no longer coupled to its endogenous localization or regulation. This result strengthens the involvement of kinesin-1 activity in QPD formation and corroborates the results observed with Kin330 transfection. Moreover, the fact that QPD formation is observed in the presence of kinesore suggests that kinesore does not compete directly with QPD-OTf binding.

**Kinesin-1 forms QPD crystals in vitro**. In a cell, multiple proteins can interact and show enzymatic activity. To pin down if kinesin-1 is the candidate to convert QPD-OTf to QPD crystals we analyzed a reconstituted in vitro system with purified proteins. We tested the crystal formation under several conditions in presence of kinesin-1, ± tubulin, MT, ATP, GTP, AMP-PNP in BRB buffer. Samples containing both kinesin-1 and MTs had a strong QPD fluorescence (Fig. 6a–c, samples 3–5), with the most intense signal deriving from the sample containing QPD-OTf, kinesin, tubulin, and GTP (sample 4). In addition, filamentous structures were observed in the MT/kinesin/QPD-OTf samples. Confocal microscopy confirmed that the filamentous-QPD structures were fluorescent (Fig. 6d). In presence of a non-hydrolysable analog of ATP (AMP-PNP), where kinesin-1 is motility is reduced while bound to MTs[42,43], we observed lower levels of fluorescent precipitate (Fig. 6a–c, sample 3). This reduced signal is consistent with our in cellulo observation where kinesin-1 motor activity is required for QPD-OTf conversion. The presence of ATP also slightly reduced the formation of the precipitate (Fig. 6a–c, sample 5). Collectively, this shows that kinesin-1 converts QPD-OTf to QPD and suggests that QPD-OTf

binding is competitive (directly or allosterically) with ATP. These results suggest a potential interaction between QPD-OTf and the ATP binding site of kinesin-1; the ATPase activity of the kinesin-1 motor domain might be serving as enzymatic activity responsible for the triflate cleavage. The fact that fluorescent crystals are observed along the filaments in the absence of ATP suggests that QPD-OTf can act as a substrate for kinesin-1.

**QPD-OTf is a substrate analog of ATP**. Taken together, the cellular and biochemical data show the dependence of crystal formation upon kinesin-1 motion on MTs. Since kinesin-1 exploits ATP hydrolysis to propel its motor domain processively on MTs[44], and that ATP is not required for crystal formation while AMP-PNP reduces crystal formation, we hypothesized that QPD-OTf acts as a substrate analog. We performed molecular docking of QPD-OTf into the ATP binding pocket of the kinesin-1 motor domain. We calculated the fitting into human kinesin-1 in the ATP state (PDB: 3J8Y) using Autodock Vina[45]. The best pose offered calculated binding energy of −8.3 kcal/mol. Superposition of this binding pose with ATP showed that the triflate overlaps with the hydrolyzed phosphate of ATP (Fig. 6e). Based on the structural similarities between QPD and ispinesib[46], an allosteric Eg5 inhibitor that also has a chloroquinozolinone moiety, we also performed docking calculations for QPD-OTf with Eg5 (PDB: 4AP0). QPD-OTf shows good pose correlation with Ispinesib (Supplementary Fig. 16), however, binding to this allosteric pocket cannot yield QPD precipitates since it positions the triflate too far from the site of hydrolysis. We next docked QPD-OTf in the nucleotide-binding site of Eg5, affording a good affinity (−8.2 kcal/mol); however, this pose positioned the triflate towards the solvent, making the triflate hydrolysis impossible (Supplementary Fig. 17). Docking studies with kinesin-1 indicated less favorable binding (−5.8 kcal/mol) in the allosteric site (Supplementary Fig. 18). Collectively, these docking studies support direct hydrolysis of the triflate of QPD-OTf and provide a rational for the selectivity of kinesin-1 over Eg5. In order to verify this putative selectivity based on the docking model with in cellulo evidence, we analyzed images of mitotic cells treated with QPD-OTf. Eg5 associates with the mitotic spindle[47,48], hence, Eg5 hydrolysis should result in fluorescence at the mitotic spindle. Imaging of mitotic HeLa-GFP-Tubulin cells treated with QPD-OTf did not show crystals emanating from the mitotic spindle but did show the expected crystals consistent with Golgi trafficking, (Supplementary Fig. 19), indicating that QPD-OTf is not a

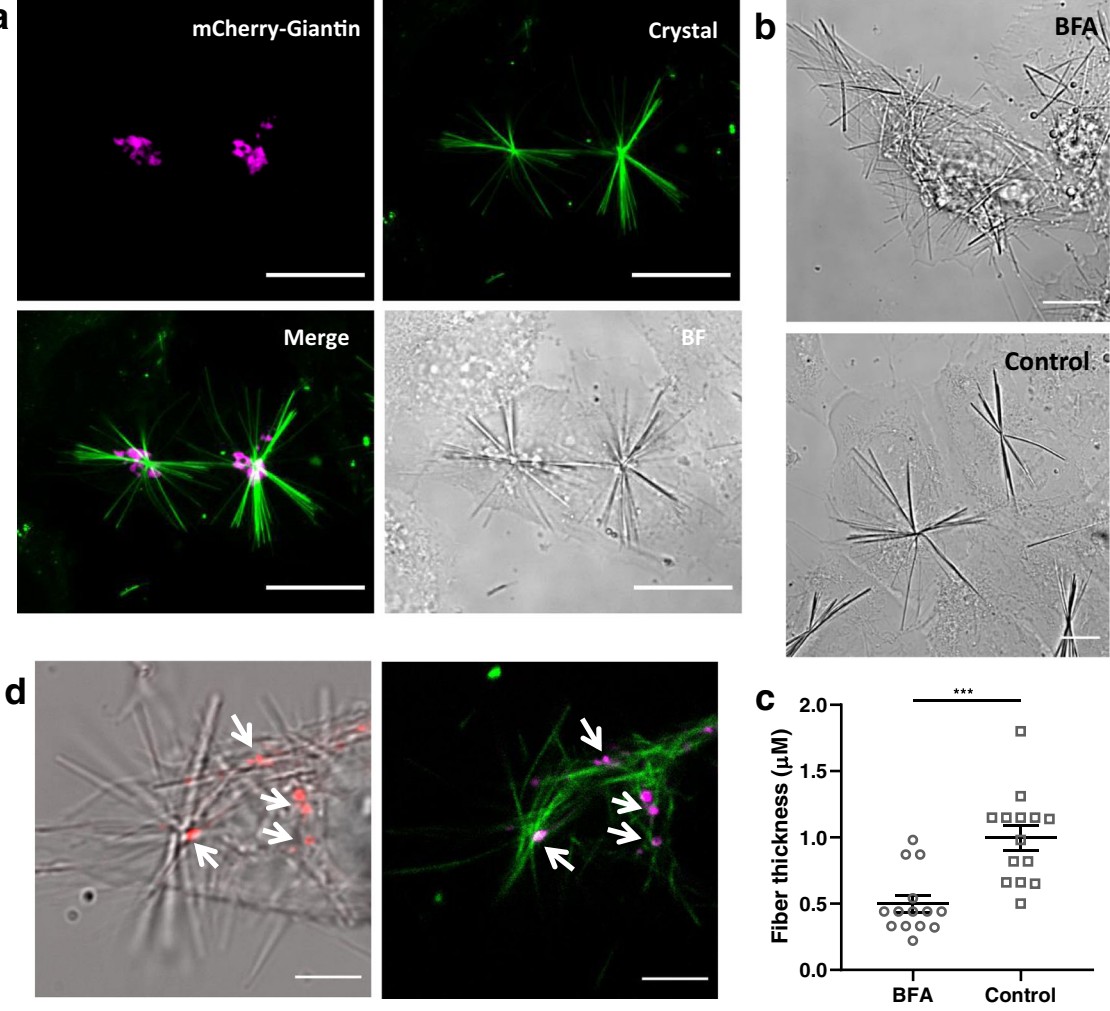

**Fig. 4 The nucleation center of the QPD crystals is localized at the Golgi apparatus. a** Representative images of mCherry-Giantin transfected U2OS cells treated with QPD-OTf (20 μM 3 h). Crystals (green), mCherry-Giantin (magenta). Scale bar 20 μM. **b** Representative images showing the effect of brefeldin A (BFA) on crystal morphology and location. BFA treated cells (20 μM BFA 4 h + 20 μM QPD-OTf 2.5 h) (top). Control (20 μM QPD-OTf 2.5 h) (bottom); scale bar 10 μM. **c** Quantification of images reported in **b**; $n = 14$ fibers; data are presented as mean value ± the standard error of the mean (SEM); statistics were calculated using a two-tailed $t$-test; ***$p = 0.0001$. **d** Localization of crystals and Golgi vesicles after BFA treatment; Golgi (red) crystals (bright field) (left); Golgi (magenta) crystals (green) (right); arrows indicate centers of crystal; scale bar 5 μM. Source data are provided as Source Data file.

substrate for Eg5. This is corroborated by the data depleting kinesin-1 using siRNA (Fig. 5d, e) that showed a ~~dramatic~~ reduction in crystal formation.

## Discussion

Small molecule fluorophore conjugates have been a powerful approach to monitor a protein of interest and the development of fluorogenic probes for live-cell imaging of the cytoskeleton, for example, have empowered cellular biology studies[31]. Alternatively, fluorogenic probes have been designed to report on a given enzymatic activity by introducing a masked fluorophore as a leaving group in an enzymatic reaction, thus acting as an activity-based fluorescent reporter[49]. While this approach has been very productive in image hydrolytic enzymes, such as protease and glycosidase, with a broad tolerance for the leaving group, there are no examples reported for motor proteins. The discovery of a fluorogenic substrate (QPD-OTf) to image kinesin-1 in live cells shows that it is possible. Moreover, the hydrolysis of a phenolic triflate represents an alternative modality for activity-based probes. This substrate is particularly attractive for a motor

protein since its fluorescent product precipitates and leaves a bright fluorescent trail along the path traveled by kinesin-1. The FIB-SEM images showed a clear helicity in the fibers, indicating that the crystals were staining a biological structure.

Until now it was difficult to record native kinesin-1 activity in cells. Kinesin-1-GFP expression at native level results in a high fluorescent background of inactive kinesin-1-GFP and it is therefore impossible to distinguish which microtubules are used for transport[19]. Complex experimental setups have been developed, like tracing microtubule dynamics in vivo, fixing cells, and adding purified tagged kinesins to map which microtubules are likely to be used for transport. Our dye shows a possibility to record native kinesin-1 activity live in a cell without any modification or fixation. The development of QPD-OTf opens the possibility to map the usage of a subset of microtubules within the dense and dynamic microtubule network.

In summary, we report an activity-based substrate for kinesin-1 yielding a bright precipitate in response to kinesin-1 activity along MTs. Based on the kinesin-1's transport activity from the Golgi, fibers are observed as a function of time, developing from foci at the Golgi. The center of the crystals reflects the location of Golgi

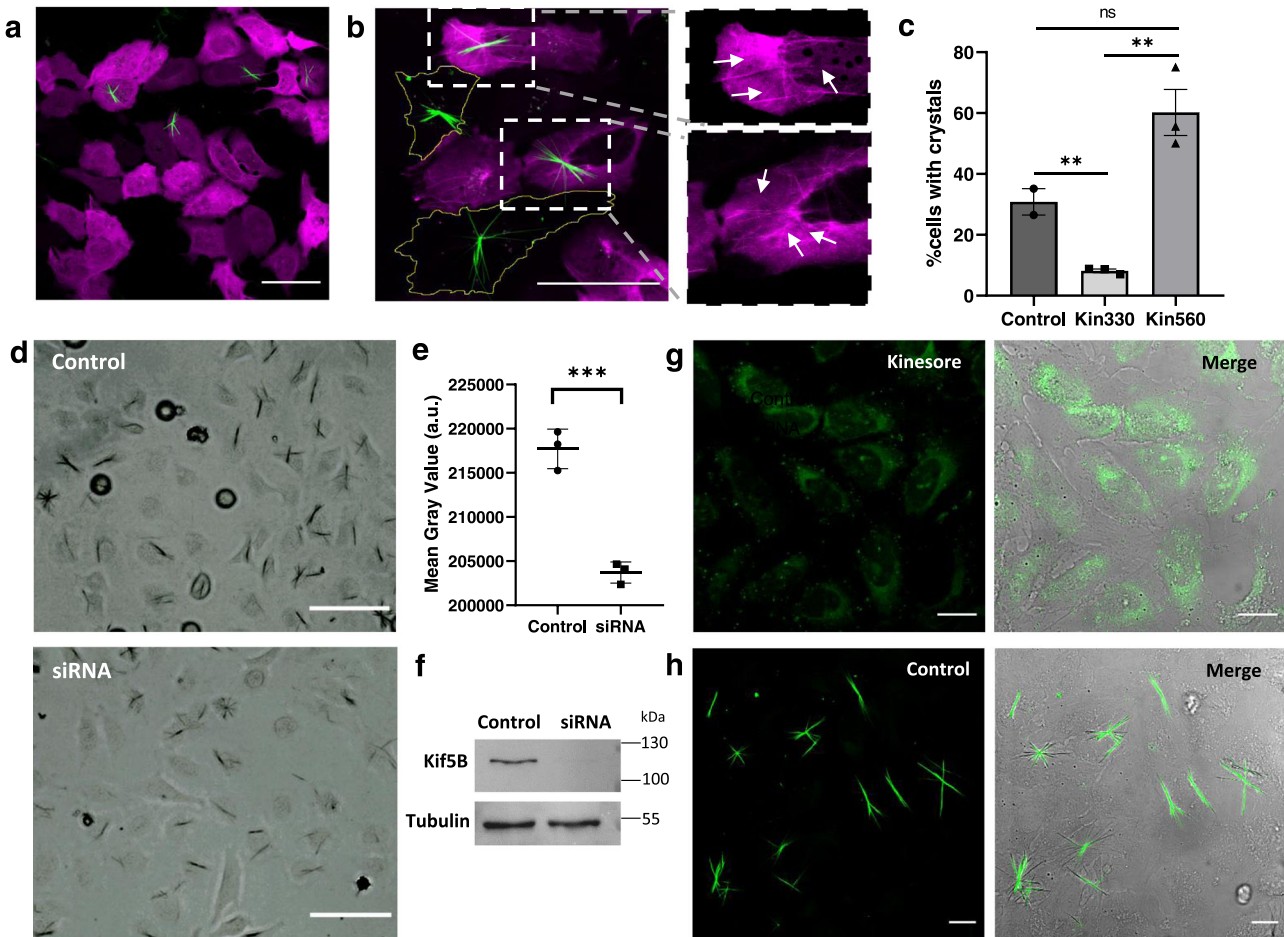

**Fig. 5 Kinesin-1 activity is required for QPD crystal formation in living cells. a** U2OS transfected with Kin330-GFP plasmid and treated with QPD-OTf (20 μM, 2.5 h); green: crystals, magenta: Kin330-GFP. **b** U2OS transfected with Kin560-GFP plasmid and treated with QPD-OTf (20 μM, 2.5 h) (left); zoom of highlighted boxes, arrows indicate stabilized MTs correlating with crystals (right); green: crystals, magenta: kinesin. Scale bar: 50 μm. **c** Quantification of crystal formation in transfected cells vs control; $n = 20$; an average of three independent experiments; data are presented as mean value ± the standard error of the mean (SEM). Statistics were calculated using a two-tailed $t$-test; **$p = 0.0064$ (Control vs Kin330), **$p = 0.0024$ (Kin330 vs Kin560); ns$p = 0.065$. **d** Kinesin-1 knockdown experiment. Representative images of HeLa-GFP-Tubulin treated with RNA control sequence + QPD-OTf (20 μM, 2 h) (top) or with kinesin-1 siRNA + QPD-OTf (20 μM, 2 h) (bottom). Scale bar 100 μm. **e** Quantification of crystal intensity for the kinesin-1 knockdown experiment. a.u. represent arbitrary units; $n = 30$; data are the average of three independent experiments; data are presented as mean value ± the standard deviation (SD); statistics were calculated using a two-tailed $t$-test; ***$p = 0.0007$. **f** Kif5B and Tubulin bands from western blot assay for the kinesin-1 knockdown experiment in HeLa cells. Samples derive from the same experiment and blots were processed in parallel. **g** U2OS treated with kinesore (100 μM) in Ringer's buffer + QPD-OTf (20 μM); green: QPD fluorescence. **h** Control conditions for experiment reported in **d** (QPD-OTf 20 μM, 2 h in Ringer's buffer); green: crystals. Scale bar 20 μm. Source data are provided as Source Data file.

elements; the number of crystals per cell and their thickness correlates with Golgi compactness/fragmentation. The crystal formation is sensitive to kinesin-1 motility; kinesin-1 depletion disrupts the formation of the crystals. In addition, the presence of MTs is required to generate QPD fluorescence in vitro. The biochemical data and docking studies support an ATP competitive mechanism involving QPD-OTf binding to the nucleotide pocket and acting as a substrate resulting in triflate hydrolysis. The resulting QPD product precipitates to form a bright fluorescent fiber along the microtubules used by kinesin-1. QPD-OTf staining is compatible with live-cell imaging; the possibility to dissolve the crystals in cell media after staining provides a non-destructive method to visualize the motion of kinesin-1 on Golgi derived MTs.

## Methods

**Cell culture**. U2OS, HeLa, HEK293T, MCF-7, RAW246.7 cell lines were obtained from the American Type Culture Collection (ATCC) and cultured according to their instructions. U2OS cells were grown in McCoy's 5A (modified) medium (Gibco) containing 10% FCS and 1% pen–strep at 37 °C under 5% $CO_2$ in a

humidified incubator. Stable expressing GFP-Tubulin Ptk2 cells (a kind gift from Franck Perez) were cultured in alpha-MEM (Gibco) containing 10% FCS and 1% pen–strep at 37 °C under 5% $CO_2$ in a humidified incubator. GFP-Tubulin CRISPR knock-in Hela cells (by C. Aumeier) were cultured in DMEM (Gibco) containing 10% FCS and 1% pen–strep at 37 °C under 5% $CO_2$ in a humidified incubator. Cells were regularly tested for mycoplasma contamination by staining with Hoechst 33342.

**Crystal formation in cells**. QPD-OTf (20 μM) was added to cells in DMEM (−) without additives and incubated at 37 °C, 5% $CO_2$ from 20 min to 4 h. Crystals can be easily detected by a ×20 objective. Super-resolution imaging was performed using adaptive deconvolution with Leica SP8 LIGHTNING with a ×63 objective.

**Live-cell imaging of QPD-OTf treated cells**. PTK2-GFP-Tubulin or HeLa-GFP-Tubulin cells ($2 \times 10^5$) were seeded into 3.5-cm-glass bottom dishes with 10 mm microwell (Mattek); cells were incubated in a culture medium at 37 °C under 5% $CO_2$ in a humidified incubator for 24 h. Then media was removed, cells were washed twice with DMEM (−) (no additives) and QPD-OTf (20 μM) was added to cells in DMEM (−) (no additives). Cells were incubated at 37 °C under 5% $CO_2$ for 2 h or for 20 min (early-stage crystals). Cells were washed twice with DMEM (−) and imaged with a LEICA SP8 microscope or with a LIGHTNING module for super-resolution images.

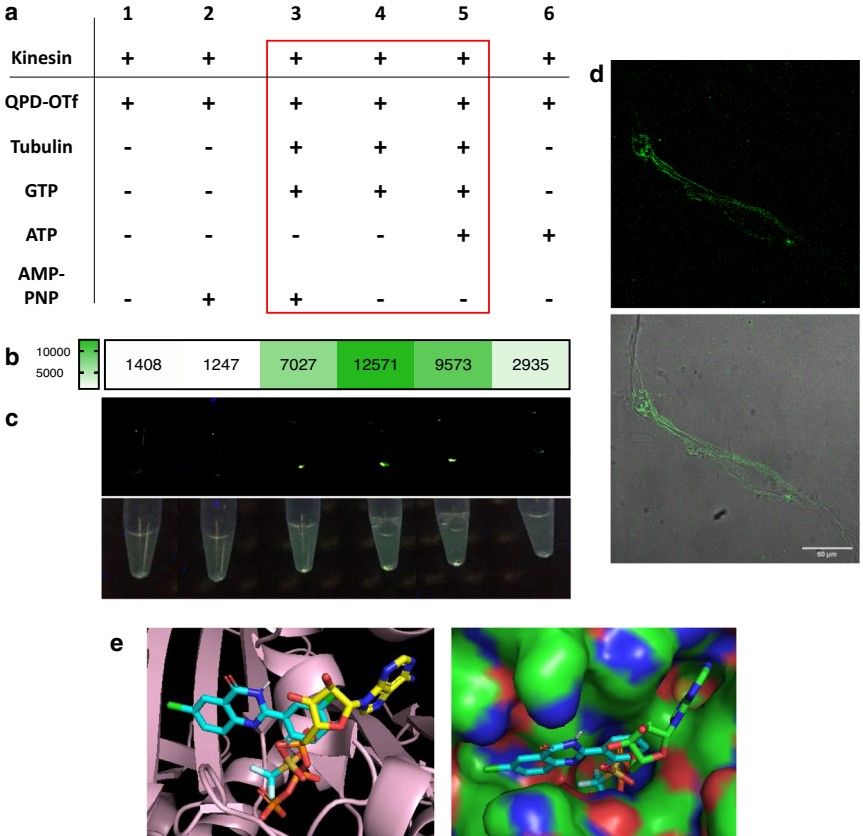

**Fig. 6 In vitro and computational studies of QPD-OTf. a** In vitro precipitation of QPD; table of conditions for the samples reported in **c**; red square indicates the samples that gave detectable QPD-fluorescence. **b** Intensity map of emitted light by samples 1–6 under 366 nm excitation; intensity values are expressed as gray values from the green channel of an RGB picture acquired with a smartphone camera. **c** Samples under 366 nm light: green channel (top); original picture (bottom). **d** Confocal imaging of fluorescent filaments contained in sample 4; green: crystal. Scale bar 50 μm. **e** Docking of QPD-OTf into ATP binding site of kinesin-1. Left: QPD-OTf (cyan), ATP (yellow), kinesin-1 (pink); right: QPD-OTf (cyan), ATP (green), kinesin-1 (polarized surface). Source data are provided as Source Data file.

**Sample preparation for FIB-SEM**. HeLa cells were grown on MatTek™ glass coverslips for 2 days. Cells were then washed × 3 with Hank's Balanced Salt solution and QPD-OTf (20 μM) in DMEM without serum was added. Cells were then incubated for 4 h at 37 °C with 5% $CO_2$. Cells were then washed with Hank's Balanced Salt solution, fixed, and processed as previously described[50] with some differences. Following dehydration, the MatTek™ glass coverslips were lifted out by partially dissolving the plastic using propylene oxide. Glass coverslips were then washed with 100% ethanol, the samples were then infiltrated with consecutively increasing concentrations of Durcupan ACM in ethanol (25:75 for 1.5 h, 50:50 for 1.5 h, 75:25 overnight). The following day the glass coverslips were immersed in 100% Durcupan ACM for 1 h, after which the resin was replaced with fresh Durcupan ACM. This was repeated four to five times. Excess Durcupan was removed using filter paper, after which the glass coverslips were heated in an oven for 10 min at 60 °C. In order to ensure a thin resin layer, the glass coverslips were centrifuged for 15 min at 37 °C and 750 RCF by placing the coverslips vertically in folded filter paper inside 50 mL Falcon tubes. The samples were then placed in an oven to polymerize at 60° under vacuum for 2 days. The coverslips were then prepared for FIB-SEM by first sputter coating with 50 nm gold, painted with silver paint, and dried under vacuum.

**FIB-SEM**. Data sets were acquired using a Zeiss Crossbeam 540 (Carl Zeiss Microscopy GmbH, Jena, Germany). Platinum and Carbon was deposited over the region of interest and the run was set up and controlled by Atlas5 software (Fibics) SEM settings: 1.5 kV; 2.5 nA; Milling probe: 300 pA. The Slice thickness and the voxel size was set to 5 nm. The total volume acquired was: $16.36 \times 9.87 \times 7.31$ μm (XYZ) and $23.5 \times 9.60 \times 7.47$ μm (XYZ).

**FIB-SEM data analysis, segmentation, and rendering**. The FIB-SEM data sets were aligned using Atlas5 software (Fibics). Data were then imported into Fiji software[2] and binned 3×, to $15 \times 15 \times 15$ nm isotropic voxels. Segmentation of structures of interest was performed using the Pixel Classification module in the Ilastik software package (Ilastik.org)[3]. The probability maps were then imported into Imaris (Bitplane.com) and surfaces were generated around fully segmented

structures. Images and videos were rendered using Imaris. Crystal cross-sections based on surface renderings were measured in Imaris.

**Fixed cells imaging of QPD-OTf-treated cells**. U2OS cells were grown in DMEM medium + 10% FBS to 50% confluency on 12-mm-glass slides (seeded the day prior). Cells were treated for 24 h with QPD-OTf (20 μM). After 24 h, cells were fixed with MeOH fixation at −20 °C for 5 min. Then the coverslips were washed for 30 min in PBS. Primary antibody staining was performed with 1:1000 dilution of DM1-alpha raised in mouse (T6199) and 1:1000 phalloidin raised in rabbit for 1 h. Coverslips were washed in PBS for 30 min. Secondary antibody staining was performed with 1:400 dilution of anti-mouse ALEXA-488 and 1:400 dilution of 1:400 anti-rabbit ALEXA-568. Coverslips were washed in PBS for 30 min. Then coverslips were placed over DABCO mounting medium containing DAPI and imaged with an LSM700 microscope.

**MTs stabilization with Taxol in live U2OS cells**. U2OS cells ($2 \times 10^5$) were seeded into 3.5-cm-glass bottom dishes with 10 mm microwell (Mattek); cells were incubated in McCoy's 5A medium at 37 °C under 5% $CO_2$ in a humidified incubator for 24 h. Then media was removed, cells were washed twice with DMEM (−) (no additives), Taxol (1 μM) was added to cells in DMEM (−) (no additives) and cells were incubated at 37 °C under 5% $CO_2$ for 1 h. QPD-OTf (20 μM) was then added and cells were incubated at 37 °C under 5% $CO_2$ for 4 h. Cells were imaged with a ×20 objective on an EVOS XL Core.

**MTs depolymerization on ice in live U2OS cells**. U2OS cells ($2 \times 10^5$) were seeded into 3.5-cm-glass bottom dishes with 10 mm microwell (Mattek); cells were incubated in McCoy's 5A medium at 37 °C under 5% $CO_2$ in a humidified incubator for 24 h. Then media was removed, cells were washed twice with DMEM (−) (no additives) and put on ice for 1 h. QPD-OTf (20 μM) was then added and cells were incubated on ice for 4 h. Cells were imaged with a ×20 objective on an EVOS XL Core. Control cells were washed with DMEM (−) (no additives) and incubated with QPD-OTf (20 μM) at 37 °C under 5% $CO_2$ for 4 h.

**Transient transfection with mCherry-Giantin plasmid**. U2OS or PTK2-GFP-Tubulin cells ($1.5 \times 10^5$) were seeded into 3.5 cm glass bottom dishes with 10 mm microwell (Mattek); cells were incubated in a culture medium at 37 °C under 5% $CO_2$ in a humidified incubator for 24 h. pSF-mCherry-SNAP-Giantin plasmid (kind gift of Riezman's lab; University of Geneva, Switzerland) was transfected with FugeneHD reagent in Optimem (100 μL); cells were incubated at 37 °C under 5% $CO_2$ for 24 h. Cells were washed twice with DMEM (−) (no additives) and QPD-OTf (20 μM) was added to cells in DMEM (−) (no additives). Cells were incubated at 37 °C under 5% $CO_2$ for 3 h. Cells were washed twice with DMEM (−) and imaged with a LEICA SP8 microscope.

**Brefeldin A treatment in live U2OS cells**. U2OS cells ($2 \times 10^5$) were seeded into 3.5 cm glass bottom dishes with 10 mm microwell (Mattek); cells were incubated in McCoy's 5A medium at 37 °C under 5% $CO_2$ in a humidified incubator overnight. Then media was removed, cells were washed twice with DMEM (−) (no additives), Brefeldin A (20 μM) was added to cells in DMEM (−) (no additives) and cells were incubated at 37 °C under 5% $CO_2$ for 4 h. Then media was replaced with fresh one containing Brefeldin A (20 μM) and QPD-OTf (20 μM) and cells were incubated at 37 °C under 5% $CO_2$ for 2.5 h. Cells were washed twice with DMEM (−) and imaged with a LEICA SP8 microscope. The same protocol was used for cells transfected with mCherry-Giantin plasmid.

**Kinesore + QPD-OTf treatment in live cells**. U2OS or PTK2-GFP-Tubulin cells ($1.5 \times 10^5$) were seeded into 3.5-cm-glass bottom dishes with 10 mm microwell (Mattek); cells were incubated in a culture medium at 37 °C under 5% $CO_2$ in a humidified incubator for 24 h. Then media was removed, cells were washed twice with DMEM (−) (no additives), Kinesore (100 μM) + QPD-OTf (20 μM) were then added to cells in Ringer's buffer and cells were incubated at 37 °C under 0% $CO_2$ for (1.5 h for PTK2; 2 h for U2OS). Cells were imaged with a LEICA SP8.

**Kinesore treatment in live cells**. U2OS cells ($1.5 \times 10^5$) were seeded into 3.5-cm-glass bottom dishes with 10 mm microwell (Mattek); cells were incubated in a culture medium at 37 °C under 5% $CO_2$ in a humidified incubator for 24 h. Then media was removed, cells were washed twice with DMEM (−) (no additives), Kinesore (100 μM) was added to cells in Ringer's buffer and cells were incubated at 37 °C under 0% $CO_2$ for 1.5 h. Cells were imaged with a LEICA SP8.

**Transient transfection with Kin330-GFP/Kin560-GFP**. U2OS cells ($1.5 \times 10^5$) were seeded into 3.5-cm-glass bottom dishes with 10 mm microwell (MatTek); cells were incubated in a culture medium at 37 °C under 5% $CO_2$ in a humidified incubator for 24 h. Kin330-GFP or Kin560-GFP plasmid was transfected with FugeneHD reagent in Opti-Mem (100 μL); cells were incubated at 37 °C under 5% $CO_2$ for 24 h. Cells were washed twice with DMEM (−) (no additives) and QPD-OTf (20 μM) was added to cells in DMEM (−) (no additives). Cells were incubated at 37 °C under 5% $CO_2$ for 3 h. Cells were washed twice with DMEM (−) and imaged with a LEICA SP8 microscope.

**Kinesin-1 knockdown**. HeLa-GFP-Tubulin cells ($7.5 \times 10^4$) were seeded into 6 well plates; cells were incubated in a culture medium at 37 °C under 5% $CO_2$ in a humidified incubator overnight. Then media was replaced with a fresh one and cells were transfected with AllStars Negative Control siRNA or with a combination of four siRNA duplexes against Kif5B subunit of kinesin-1 (GeneSolution siRNA, Qiagen) at a final concentration of 10 nM in Lipofectamine RNAiMAX. Cells were incubated at 37 °C under 5% $CO_2$ in a humidified incubator for 72 h. Cells were washed with DMEM (−) (no additives), incubated with QPD-OTf (20 μM), and imaged after 2 h.

**In vitro precipitation of QPD**. 20 μM QPD was precipitated in Eppendorf tubes at room temperature for 6 h in BRB80 in presence of different combinations of unlabeled 14 μM tubulin, 150 nM kinesin-1, 2.7 mM AMP-PNP, 2.7 mM ATP, 1 mM GTP. Samples were visualized under a 366 nm lamp. The content of samples containing fluorescent precipitate was imaged by a LEICA SP8 microscope.

**Western blot**. To HeLa-GFP-Tubulin transfected with siRNA was added lysis buffer and cells were let 5 min on ice before being scraped. The cell lysate was transferred into Eppendorf tubes and incubated on ice for 30 min, and then centrifuged at 14,000×$g$ at 4 °C for 20 min. An aliquot of the supernatant was mixed with Laemli buffer (5×) and loaded on 8% acrylamide gel. Proteins were transferred onto PVDF membrane and blocked in 5% BDA in TBST buffer for 1 h at r.t. The membrane was incubated with anti-UKHC (kinesin) primary antibody (1: 1000) or anti-α-tubulin primary antibody (1:1000) in 5% dehydrated milk in TBST buffer at 4 °C overnight. The membrane was washed with TBST buffer (3 × 10 min) and incubated with secondary HRP antibody (1:10,000) in 5% dehydrated milk in TBST buffer for 1 h at r.t. The membrane was washed with TBST buffer (3 × 10 min). The membrane was then rinsed several times with a mixture of peroxide/luminol solution and the chemiluminescent signal was acquired. The uncropped version of the blots is provided in the Source Data file.

**Molecular docking**. Docking calculations were performed with Autodock4 Vina. Receptor (PDB structure: 3J8Y for kinesin-1, 4AP0 for Eg5) and ligand preparation were performed in AutodockTools1.5.6. Results were displayed with PyMOL2.

**Statistics and reproducibility**. All microscopy experiments were repeated at least three times with similar results.

**Reporting summary**. Further information on research design is available in the Nature Research Reporting Summary linked to this article.

## Data availability

The authors declare that all data supporting the findings of this study are available within the article and its supplementary information files. Source data are provided with this paper. The data sets generated during and/or analyzed during the current study, together with the Source Data have been deposited in the Zenodo repository (https://doi.org/10.5281/zenodo.4461867).

## Code availability

The codes generated during the current study have been deposited in the Zenodo repository (https://doi.org/10.5281/zenodo.4461867) Autodock4 Vina is an open-source program for molecular docking designed and implemented by Dr. Oleg Trott in the Molecular Graphics Lab at The Scripps Research Institute.

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

## Acknowledgements

We thank Howard Riezman's group for providing the mCherry-Giantin plasmid. We thank Paul Guichard's group for providing useful reagents and for constructive criticism of the manuscript. We thank the BioImaging Centre of the University of Geneva and ACCESS for technical assistance with super-resolution imaging. This work was supported by the NCCR chemical biology and the Department de l'instruction publique (DIP), Geneva.

## Author contributions

E.L. discovered QPD-OTf and performed initial experiments in cellulo. S.A. performed or contributed to all experiments. E.L. and C.B. performed FIB-SEM experiments. N.K. performed immunostaining experiments. C.A. performed TIRF microscopy and in vitro experiments. S.A., E.L., N.K., C.A., and N.W. designed experiments. N.W. and C.A. supervised the project. S.A., C.A., and N.W. wrote the paper. All the authors discussed and commented on the manuscript.

## Competing interests

The authors declare no competing interests.
