## [Peer Review File · Nature Communications]

Reviewers' Comments:

Reviewer #1:

Remarks to the Author:

The present manuscript by Aumeier and Winssinger et. al. reported an activity-based probe to track kinesin-1 movement in living cells. Unlike previous triflated fluorophores for detecting superoxide, this novel QPD-OTf could form a precipitating dye QPD upon enzymatic conversion of kinesin-1, giving fluorescent tracks for this motor protein along microtubules. The authors conducted a series of experiments to examine the subcellular locations of formed crystals and the effects of various drugs, including Taxol, BFA, Kin560 and Kin330 transfection. Moreover, in vitro studies and computational modeling were carried out to confirm that QPD-OTf is a substrate analog for kinesin-1 activity. This study is novel, important, and comprehensive to be published after the authors address the following concerns.

1. The reaction site of the QPD-OTf should be the motor domains of kinesin-1, and the dye crystal should grow along the direction of kinesin-1 processive movement on microtubules. In all those figures, only the static images of final crystals were presented. Did the author observe the crystal growth? What is the optimal time window to observe the crystal formation? Would it be possible to present a time-lapse dynamic image for crystal formation (e.g., 0.5, 1.0, 1.5, 2.0, 2.5 h) in living cells? This time-lapse image could further improve dynamic applications of QPD-OTf.
2. The data presented in Figs. 3 & 5 could effectively support the proposed roles of microtubules and kinesin-1 during crystal formation. However, the number of 'n', and the definition of data significance in bar charts should be clearly stated in the figure caption.
3. The current Fig. 1 only showed the 'enzymatic conversion' as the reaction mechanism, which is too ambiguous. I would suggest providing more information to help the readers to grasp the key information quickly in Fig. 1.
4. If kinesin-1 is the enzyme that hydrolyzes QPD-OTf substrate, enzyme kinetics parameters should provide more information.
5. In Figure 5 B, some crystals formed outside the cell. It means the crystals can be formed outside the cell or the crystals destroyed the cell in that position?
6. The authors said that QPD-OTf staining is compatible with live cell imaging and it is possible to dissolve the crystals in cell media after staining. If this is true, it will largely broaden the use of QPD-OTf. However, the authors only used a curve chart in Supplementary figure 3 to support it. The authors can repeat the QPD-OTf staining in the same cell several times to see the influence of QPD-OTf staining in live cells.
7. In Figure 4, the crystals grew in all directions and formed a cluster; however, in Figures 2&5, most crystals grew in opposite directions and formed a rod. Is this phenomenon just an accident or any stories behind this phenomenon?

Reviewer #2:

Remarks to the Author:

In this interesting study, Angerani and colleagues perform an in vitro biochemical and cellular characterisation of novel compound that forms striking aster-like filamentous fluorescent crystal structures in a variety of cell types. They provide evidence that these structures are (in part) associated with the microtubule network and that their formation is dependent upon it. The filaments appear to predominantly originate from the Golgi. Mechanistically, perturbation of kinesin-1 function through over-expression of functional and non-functional fragments, or by use of a kinesin-1 targeting small-molecule, modifies filament/crystal formation. In vitro, kinesin and MTs are both necessary for the generation of fluorescent (filamentous) precipitate in a simple in vitro reconstitution system. A series computational docking analyses suggest that the compound may engage the ATP binding pocket of kinesin-1 and be processed there.

The authors conclude that the novel compound provides the capacity to trace kinesin-1 motility i.e. that the crystals/filaments provide a record of kinesin-1 activity and that this is localised to specific

microtubules. If so, this would be a very useful new tool giving for the first time to capacity to visualize the activity of endogenous kinesin-1 for the first time.

I am just about convinced that these structures are formed through a microtubule-associated enzymatic activity and that this probably involves a kinesin family member. In and of itself, that is an interesting finding worth reporting. Unfortunately, the data provided are insufficient to support the broader argument and main points of the manuscript that this compound reports on kinesin-1 specifically (as opposed to one or more of the many other kinesins), nor that the crystals/filaments reflect specific motor-microtubule transport paths.

Areas that need substantial further work include the fluorescence imaging showing association of the filaments/crystals with microtubules which seems to vary in quality and morphology hugely across the manuscript; a more careful investigation of the kinesin-1 dependence of these structures; and some higher spatial and temporal resolution analysis of their dynamics through live-cell imaging at much earlier time points. Some thoughts on how to approach this are described below.

Major comments

1. Kinesin-1 dependence of the crystal formation in cells. The authors provide evidence in vitro that kinesin activity can generate the fluorescent precipitate and some relevant manipulations of kinesin-1 in cells that suggest that it is involved in crystal formation. However, in my opinion a key experiment is missing – endogenous kinesin-1 should be depleted and the effect on crystal formation should be determined. I suggest that the most straightforward way to do this is using siRNA directed against Kif5B (the main heavy chain paralogue in HeLa cells). Alternatively, several labs (Ahkmanova or Bonifacino) have published CRISPR knockouts of Kif5B so it could be possible to request these. In either case, this should be provided with a western blot showing efficient depletion and quantification of crystal formation.

2. Imaging of crystals and microtubules. There appears to be a huge variation in both the quality of the fluorescence staining/imaging throughout the manuscript and the morphology of the crystals/filaments. This makes it very difficult for the reader to understand what is representative and to properly evaluate the data. For example, Figure 1C and 1D show aster-like crystals and two different tubulin labels (SiR-tubulin and GFP-tubulin), neither of which looks very much like microtubules in U2OS cells. In contrast, figure 1B shows reasonable tubulin antibody staining but with crystals of a very different morphology where it is difficult to conclude that they are microtubule-associated. As such, it is frustrating to be left with some doubt over one of the most basic points of the manuscript: that these structures are substantially microtubule associated and not simply crystals that are growing from a nucleation point that is typically in the centre of the cell. I suggest that without substantial optimisation the SiR-tubulin and GFP-tubulin are of limited value here and that the authors should focus on getting high quality, high resolution images using tubulin immunostaining (beta-tub antibodies with methanol fixation are usually the best) to provide really robust evidence that the crystals are occurring along microtubules. Also, I can't see any evidence of juxtaposed microtubules in the EM provided. This may be for technical reasons but if the authors have it, I'd encourage them to present it.

3. Live imaging and high spatial and temporal resolution. If I understand the authors proposition correctly, kinesin-1 running on microtubules generates QPD precipitates that presumably remain localised close to the site of generation. These crystallise as the concentration increases. These are very large (100 nm – 700 nm diameter) crystals structure that are highly rigid and so can deform both the nuclear envelope and plasma membrane. These structures are clearly many times thicker than the microtubules/kinesin traces they are supposedly marking (25 nm diameter microtubule). If so, to me, this seems to be a highly artifactual end stage and not very useful for understanding kinesin biology as the paper suggests. Of much more interest is the initial deposition of QPD at

very early timepoints, imaged at high- or perhaps super- resolution, ideally live so that the initial deposition of fluorescent material along microtubule tracks can be visualised. I think this is crucial to support the main claim of the manuscript that kinesin-1 motility is 'traced'. Perhaps this might also provide for more convincing co-localisation of QPD and microtubules as per point 2.

Additional points

4. The background and discussion on kinesin-1 is quite limited. The introduction should refer to the fact that 'kinesin-1' it is a family of closely related tetrameric enzymes and describe their subunit composition. This is important because the authors go on to use specific heavy chain fragments that incorporate the motor domain and it is important for the reader to understand what these are.

5. The section 'Purified microtubules are not sufficient to generate crystals in vitro' refers to a TIRF assay. There doesn't seem to be any data associated with this section although there is a section in the supplementary methods describing the protocol. Its ok to say that some avenues were explored without success as part of the narrative, but if the authors wish to report a negative result : "No crystal formation could be observed and no fluorescence of QPD was detected in our in vitro TIRF assay, even after 2 hours" they should provide the data and with appropriate controls.

6. In general, the figure legends lack detail. How many independent experiments or replicates were performed. What are the sample sizes? What statistical tests were used for data analysis?

7. There seems to be some over-interpretation in the docking experiments. It is an interesting model and should be presented, but a lack of orthogonal experimental data prevent conclusions such as 'Collectively, these docking studies support a direct hydrolysis of the triflate of QPD-OTf and provide a rationale for the selectivity of kinesin-1 over Eg5 and kin-1'. Not least, I see no evidence anywhere in the manuscript for selectivity over Eg5 (this would need Eg5 to be incorporated into the in vitro assays at a minimum) and I am not sure what kin-1 is in this context (C.elegans homologue?).

8. To support their arguments, the authors might consider 'redirecting' bulk kinesin-1 activity away from the Golgi to ask whether the organisation of the crystals changes. This could be done using the Arl8/SKIP overexpression system (Rosa-Ferreira and Munro, Dev Cell, 2011) to enhance recruitment to lysosomes.

Reviewer #3:

Remarks to the Author:

In the manuscript entitled "Kinesin-1 motility traced by an activity-based precipitating dye" by Angerani et al., the authors reported the discovery of a fluorogenic substrate (QPD-OTf) for tracing the activity of kinesin-1 in vivo. Importantly, the authors showed that the substrate fluorogenic substrate acts as an ATP analogue and binds to Kinesin-1 to yield an insoluble fluorescent dye that decorates on the path traveled by Kinesin-1. Overall, this is a very interesting work and potentially significant for the kinesin field. Before this manuscript can be considered for publication, the following comments need to be sufficiently addressed.

1. The Abstract in the present form does not concisely summarize the major findings of the work and the implication(s) of these findings. The authors should consider rewriting a new Abstract by combining Lines 14-18 (of the current Abstract) and the majority of the last paragraph of the Introduction (Lines 51-60).

2. Captions of many figures need to be revised to make them more informative, which will help the readers better appreciate the results. For some figures, the panels need to be reordered too.

For Figure 1, change the title to "Schema of the soluble profluorophore QPD-OTf and the insoluble fluorescent dye QPD", and remove the panel on the right.

For Figure 2, change the caption to "QPD-OTf forms crystals that strongly colocalize with MTs in living cells."

For Figure 3, change the title to "Formation of QPD crystals in U2OS live cells is disrupted by induced microtubule stabilization or depolymerization". For Fig. 3A, the authors should consider writing "Representative images of crystal formation in cells treated (left) with 1 μ M Taxol for 1 hour and 20 μ M QPD-OTf for 4 hours at 37 C; (middle) on ice for 1 hour and with 20 μ M QPD-OTf for 4 hours on ice; and (right) with 20 μ M QPD-OTf for 4 hours at 37 C. (Bottom) Zoomed-in images of cells in the black squares". To better help the readers comprehend Fig. 2A, the authors should use dashed lines to link the original boxes and the corresponding zoomed-in images.

For Figure 4, change the title to "The nucleation center of the QPD crystals is localized at the Golgi apparatus". Similar to Fig. 3A, the text for Fig. 4A should be revised to make it more descriptive and informative for the readers.

For Figure 5, change the title to "Kinesin-1 activity is required for QPD crystal formation in vivo". The authors should also add the control images for Figs. 5A and B, and add dashed lines to link the white boxes and the corresponding zoomed-in images in Fig. 5B.

For Figure 6, I suggest the authors to swap Fig. 6A and Fig. 6C so that the table of conditions for the 6 in vitro precipitation experiments of QPD is presented first, followed by the images of the 6 samples and the intensity readouts of all 6 samples.

For Supplementary Figure 6, change the title to "The centrosome does not exhibit colocalization with the nucleating site for QPD precipitation".

3. In Lines 96-99, the authors need to consider revising the text to "We first treated U2OS cells with 1 μ M Taxol and 20 μ M QPD-OTf for 4 hours and found that compared to the control experiments (Fig. 3A, right), Taxol-induced MT stabilization reduced the crystal formation by 75% (Fig. 3A, left; and Fig. 3B). I suggest the authors to re-order the image sequence to "Control", "Taxol" and "On ice" (from left to right).

4. In Line 101, please clearly indicate that the middle panel of Fig. 3A is the image that corresponds to the statement "no crystals were observed (Fig. 3)".

5. In Line 115, please clarify the operation of sequential treatment of QPD-OTf. Did the authors mean "subsequent" treatment of QPD-DTF?

6. In Fig. 5C, the authors referred to the two kinesin constructs as Kin560 and Kin330, but in all other places of the manuscript, these two constructs were referred to kin330 (Lines 141, 159 and 466) and kin560 (Lines 141, 146-148 and 467). To be consistent and avoid confusion, the authors need to use the same naming convention for both constructs throughout the manuscript.

7. In Lines 171-173, Fig. 8A-C should be Fig. 6A-C.

8. In Line 179, change the subsection heading to "QPD-OTf is a substrate analogue of ATP."

9. In the Section entitled "QPD-OTf conversion to QPD depends on kinesin-1 motility" (Lines 137-161), several important observations were made that are key to understand the role of kinesin-1 activity in the conversion of QPD-OTf to QPD. The authors need to expand this section to explain: 1) why transfection with the immotile Kin330 reduced the number of crystals by 87% compared

with the non-transfected control cells; and 2) why activation of kinesin-1 with kinesore does not mimick transfection with Kin560 in terms of crystal formation.

10. The Discussion needs to be significantly improved to help the readers better appreciate the results and the underlying mechanisms and the significance and implication(s) of this work. I suggest that authors to: 1) move up the last paragraph of the Discussion to become the starting paragraph; 2) add one or two paragraphs to discuss about the molecular mechanisms of some key observations that may not be immediately clear to the readers; and 3) add at least one more paragraph to discuss how others can take advantage of the fluorogenic substrate in studying kinesin-1 activity in vivo and how similar fluorogenic substrates specific for other kinesin motors such as kinesin-5 can be rationally designed.

11. In Lines 219-220, the authors stated that "kinesin-1 inhibitors disrupt the formation of the crystals." Are the authors referring to published work by others or results in this manuscript?

Point by point response to the reviewers' comments - NCOMMS-20-30449

Reviewer #1 (Remarks to the Author):

The present manuscript by Aumeier and Winssinger et. al. reported an activity-based probe to track kinesin-1 movement in living cells. Unlike previous triflated fluorophores for detecting superoxide, this novel QPD-OTf could form a precipitating dye QPD upon enzymatic conversion of kinesin-1, giving fluorescent tracks for this motor protein along microtubules. The authors conducted a series of experiments to examine the subcellular locations of formed crystals and the effects of various drugs, including Taxol, BFA, Kin560 and Kin330 transfection. Moreover, in vitro studies and computational modeling were carried out to confirm that QPD-OTf is a substrate analog for kinesin-1 activity. This study is novel, important, and comprehensive to be published after the authors address the following concerns.

1. The reaction site of the QPD-OTf should be the motor domains of kinesin-1, and the dye crystal should grow along the direction of kinesin-1 processive movement on microtubules. In all those figures, only the static images of final crystals were presented. Did the author observe the crystal growth? What is the optimal time window to observe the crystal formation? Would it be possible to present a time-lapse dynamic image for crystal formation (e.g., 0.5, 1.0, 1.5, 2.0, 2.5 h) in living cells? This time-lapse image could further improve dynamic applications of QPD-OTf.

This is an excellent point. A time lapse of crystal formation is now added (Supplementary movie 1). For convenience, snapshots of the time laps are also shown in Supplementary Fig. 2 for crystal formation in live PTK2-GFP-Tubulin cells as well as a time-lapse image of U2OS-mCherry-Giantin cells (Supplementary Fig 11). It should be noted that excitation using blue laser excitation has inherent limitation due to the toxicity of prolonged laser exposure limiting the number of frame that can be acquired in monitoring a dynamic process.

The optimal time window to observe crystal formation ranges from 30 min to 4 hours. We observed variation in crystal appearance depending on the cell line; HeLa and PTK2 cells develop crystal in a shorter timeframe (around 30 min); U2OS and MCF-7 cells require around 2 hours.

2. The data presented in Figs. 3 & 5 could effectively support the proposed roles of microtubules and kinesin-1 during crystal formation. However, the number of 'n', and the definition of data significance in bar charts should be clearly stated in the figure caption.

The number of "n" and data significance have now been added to the data shown in Figs. 3 & 5.

3. The current Fig. 1 only showed the 'enzymatic conversion' as the reaction mechanism, which is too ambiguous. I would suggest providing more information to help the readers to grasp the key information quickly in Fig. 1.

Fig.1 has now been modified to include kinesin-1 as trigger of the precipitation.

4. If kinesin-1 is the enzyme that hydrolyzes QPD-OTf substrate, enzyme kinetics parameters should provide more information.

This point is well taken. Kinetic measurements are unfortunately complicated by the fact that the product precipitates and measurements in microtiter plates or cuvettes are not accurate since these instruments sample only a fraction of the solution. In trying to surmount this technical problem, we have tried to quantify the amount of product formed using high throughput microscopy as a function of time across a range of substrate concentrations in order to calculate a K_m . However, the quantification is biased by crystals forming outside a single Z plane and the data is not sufficiently robust to draw rigorous kinetic measurements. From the cell-based experiments, addition of 10 or 20 μM solution of QPD-OTf does not result in significant differences in fiber formation suggesting these concentrations are not limiting the kinetics (i.e close to V_{max}). Thus, a $K_m < 5 \mu\text{M}$ can be anticipated however, this is too speculative to be included in the main text. From a practical standpoint, 10-20 μM of QPD-OTf is not toxic and insures maximum fiber formation.

5. In Figure 5 B, some crystals formed outside the cell. It means the crystals can be formed outside the cell or the crystals destroyed the cell in that position?

Crystals are never formed outside the cells. In Fig 5B, cells are apparent from the imaging of kinesin-GFP transfected. The transfection however was heterogeneous and some cells do not express kinesin-GFP but still form crystals, hence the appearance of crystals outside the cells. For clarity, the contour of cells lacking kinesin-GFP expression has now been added to Fig 5B. Bright field images of pictures displayed in Fig.5B are in Supplementary Fig. 15. These images clearly show that crystals are within cells.

6. The authors said that QPD-OTf staining is compatible with live cell imaging and it is possible to dissolve the crystals in cell media after staining. If this is true, it will largely broaden the use of QPD-OTf. However, the authors only used a curve chart in Supplementary figure 3 to support it. The authors can repeat the QPD-OTf staining in the same cell several times to see the influence of QPD-OTf staining in live cells.

This point is well taken. A time laps movie (Supplementary movie 2) is now added and snapshot showing crystals dissolving over time is now added in supplementary Fig. 5. The text has been amended to include reference to this additional Fig and movie.

7. In Figure 4, the crystals grew in all directions and formed a cluster; however, in Figures 2&5, most crystals grew in opposite directions and formed a rod. Is this phenomenon just an accident or any stories behind this phenomenon?

This is a good point. We observed slight changes in crystal morphology depending on cell type. We attribute this phenomenon to different kinesin dynamics and cell morphology among different cell lines. The following sentence has now been added in the main text: "While different cell lines afforded slightly different crystal morphology, all cell lines showed fibers that emanate from central points"

Reviewer #2 (Remarks to the Author):

In this interesting study, Angerani and colleagues perform an in vitro biochemical and cellular characterisation of novel compound that forms striking aster-like filamentous fluorescent crystal structures in a variety of cell types. They provide evidence that these structures are (in part) associated with the microtubule network and that their formation is dependent upon it. The filaments appear to predominantly originate from the Golgi. Mechanistically, perturbation of kinesin-1 function through over-expression of functional and non-functional fragments, or by use of a kinesin-1 targeting small-molecule, modifies filament/crystal formation. In vitro, kinesin and MTs are both necessary for the generation of fluorescent (filamentous) precipitate in a simple in vitro reconstitution system. A series computational docking analyses suggest that the compound may engage the ATP binding pocket of kinesin-1 and be processed there.

The authors conclude that the novel compound provides the capacity to trace kinesin-1 motility i.e. that the crystals/filaments provide a record of kinesin-1 activity and that this is localised to specific microtubules. If so, this would be a very useful new tool giving for the first time to capacity to visualize the activity of endogenous kinesin-1 for the first time.

I am just about convinced that these structures are formed through a microtubule-associated enzymatic activity and that this probably involves a kinesin family member. In and of itself, that is an interesting finding worth reporting. Unfortunately, the data provided are insufficient to support the broader argument and main points of the manuscript that this compound reports on kinesin-1 specifically (as opposed to one or more of the many other kinesins), nor that the crystals/filaments reflect specific motor-microtubule transport paths.

Areas that need substantial further work include the fluorescence imaging showing association of the filaments/crystals with microtubules which seems to vary in quality and morphology hugely across the manuscript; a more careful investigation of the kinesin-1 dependence of these structures; and some higher spatial and temporal resolution analysis of their dynamics through live-cell imaging at much earlier time points. Some thoughts on how to approach this are described below.

Major comments

1. Kinesin-1 dependence of the crystal formation in cells. The authors provide evidence in vitro that

kinesin activity can generate the fluorescent precipitate and some relevant manipulations of kinesin-1 in cells that suggest that it is involved in crystal formation. However, in my opinion a key experiment is missing – endogenous kinesin-1 should be depleted and the effect on crystal formation should be determined. I suggest that the most straightforward way to do this is using siRNA directed against Kif5B (the main heavy chain paralogue in HeLa cells). Alternatively, several labs (Ahkmanova or Bonifacino) have published CRISPR knockouts of Kif5B so it could be possible to request these. In either case, this should be provided with a western blot showing efficient depletion and quantification of crystal formation.

This is an excellent suggestion. The suggested experiment has now been performed, siRNA knock down of kinesin-1 indeed showed a dramatic reduction in fiber formation. This data is shown in Fig 5D and quantified in 5E. A discussion has also been added to the main text. This experiment also added an important evidence regarding the selective reporting of kinesin-1 activity vs other motor protein (eg. Eg5)

2. Imaging of crystals and microtubules. There appears to be a huge variation in both the quality of the fluorescence staining/imaging throughout the manuscript and the morphology of the crystals/filaments. This makes it very difficult for the reader to understand what is representative and to properly evaluate the data. For example, Figure 1C and 1D show aster-like crystals and two different tubulin labels (SiR-tubulin and GFP-tubulin), neither of which looks very much like microtubules in U2OS cells. In contrast, figure 1B shows reasonable tubulin antibody staining but with crystals of a very different morphology where it is difficult to conclude that they are microtubule-associated. As such, it is frustrating to be left with some doubt over one of the most basic points of the manuscript: that these structures are substantially microtubule associated and not simply crystals that are growing from a nucleation point that is typically in the centre of the cell. I suggest that without substantial optimisation the SiR-tubulin and GFP-tubulin are of limited value here and that the authors should focus on getting high quality, high resolution images using tubulin immunostaining (beta-tub antibodies with methanol fixation are usually the best) to provide really robust evidence that the crystals are occurring along microtubules. Also, I can't see any evidence of juxtaposed microtubules in the EM provided. This may be for technical reasons but if the authors have it, I'd encourage them to present it.

This point touches one the same elements as raised by reviewer 1 (point 7). Indeed, there are different crystal morphology across different cell lines. We have now highlighted this fact in the results section. The second question is the evidence for the association of crystals with tubulin. We used three different techniques: SiR-tubulin, which has the advantage that it can be used on any cells but does not have the resolution of the second two techniques; immunostaining, which again can be used on any cell line but requires fixation and several washing steps which can compromise the crystal (redissolving or moving); a cell line that stably expresses GFP-tubulin, which give the clearest information but is confined to the availability of such cell line. We agree that the different methods give results of different quality but all the method concur with an association of the crystals and MT. We believe that presenting the different methods is helpful in offering a range of techniques, with the caveat and limitations discussed above. Super-resolution images have now been acquired (adaptive deconvolution using Leica's Lightning, 120 nm resolution) on live cells using HeLa expressing GFP-tubulin showing the colocalization of the crystal fibers a MTs (Fig 2, panel C and D). The FIB imaging is a negative stain and crystals are dissolved in the sample preparation. NB: The reviewer's comment refers to Fig 1 but based on the staining discuss, we interpret the comments as being on Fig 2 of the submitted manuscript.

3. Live imaging and high spatial and temporal resolution. If I understand the authors proposition correctly, kinesin-1 running on microtubules generates QPD precipitates that presumably remain localised close to the site of generation. These crystallise as the concentration increases. These are very large (100 nm – 700 nm diameter) crystals structure that are highly rigid and so can deform both the nuclear envelope and plasma membrane. These structures are clearly many times thicker than the microtubules/kinesin traces they are supposedly marking (25 nm diameter microtubule). If so, to me, this seems to be a highly artifactual end stage and not very useful for understanding kinesin biology as the paper suggests. Of much more interest is the initial deposition of QPD at very early timepoints, imaged at high- or perhaps super- resolution, ideally live so that the initial deposition of fluorescent material along microtubule tracks can be visualised. I think this is crucial to

support the main claim of the manuscript that kinesin-1 motility is 'traced'. Perhaps this might also provide for more convincing co-localisation of QPD and microtubules as per point 2.

The size (100-700 nm diameter) was calculated from the FIB-SEM 3D reconstruction which is acquired by negative stain using cells that were allowed to form crystals until a fairly late time point to ensure that crystal would be observed in the field of view. Based on the fact that this is a negative stain and that the crystals were dissolved, it can be concluded that the size measurements represent a maximum value that may overestimate the size of the crystals. It is clear that at this time point, MT bundle into super structure that indeed become fairly wide structure (700 nm).

The point of high resolution images is well taken and super-resolution images on live cells have now been added to Fig 2. Panel 2D shows images at early time point with crystal formation starting along single MT.

Additional points

4. The background and discussion on kinesin-1 is quite limited. The introduction should refer to the fact that 'kinesin-1' it is a family of closely related tetrameric enzymes and describe their subunit composition. This is important because the authors go on to use specific heavy chain fragments that incorporate the motor domain and it is important for the reader to understand what these are.

The introduction has now been updated and additional references have been added.

5. The section 'Purified microtubules are not sufficient to generate crystals in vitro' refers to a TIRF assay. There doesn't seem to be any data associated with this section although there is a section in the supplementary methods describing the protocol. Its ok to say that some avenues were explored without success as part of the narrative, but if the authors wish to report a negative result : "No crystal formation could be observed and no fluorescence of QPD was detected in our in vitro TIRF assay, even after 2 hours" they should provide the data and with appropriate controls.

We apologise for the missing data. TIRF images have now been included in Supplementary Fig. 14.

6. In general, the figure legends lack detail. How many independent experiments or replicates were performed. What are the sample sizes? What statistical tests were used for data analysis?

The figure legends have been updated and now include sample size, number of replicates, statistical tests.

7. There seems to be some over-interpretation in the docking experiments. It is an interesting model and should be presented, but a lack of orthogonal experimental data prevent conclusions such as 'Collectively, these docking studies support a direct hydrolysis of the triflate of QPD-OTf and provide a rationale for the selectivity of kinesin-1 over Eg5 and kin-1'. Not least, I see no evidence anywhere in the manuscript for selectivity over Eg5 (this would need Eg5 to be incorporated into the in vitro assays at a minimum) and I am not sure what kin-1 is in this context (C.elegans homologue?).

This is an excellent point; the docking experiment is only a suggestion that must be substantiated with experimental evidence. We have now added two lines of experimental evidence, 1. The siRNA knockdown of kinesin-1 led to dramatic reduction in crystal formation suggesting that Eg5 does not contribute significantly (Fig. 5D, E); 2. Analysis of cells undergoing cellular division do not show crystal formation at the mitotic spindle (Supplementary Fig. 19), where Eg5 shows a high activity. The results section has been amended to reflect these points:

"In order to verify this putative selectivity based on docking model with in cellulo evidence, we analyzed images of mitotic cells treated with QPD-OTf. Eg5 associates with the mitotic spindle^{42 43} hence, an Eg5 hydrolysis should result in fluorescence at the mitotic spindle. Imaging of mitotic HeLa-GFP-Tubulin cells treated with QPD-OTf did not show crystals emanating from the mitotic spindle but did show the expected crystals consistent with Golgi trafficking, (Supplementary Fig. 19), indicating that QPD-OTf is not a substrate for Eg5. This is corroborated by the data depleting kinesin-1 using siRNA (Fig. 5D-E) that showed a dramatic reduction in crystal formation"

"kin-1" was mistakenly used as an abbreviation of kinesin-1. This has now been corrected and the paragraph was reworked for clarity, adding the pdb number of Eg5 and distinguishing docking studies in the nucleotide binding site and allosteric binding site.

8. To support their arguments, the authors might consider 'redirecting' bulk kinesin-1 activity away from the Golgi to ask whether the organisation of the crystals changes. This could be done using the Arl8/SKIP overexpression system (Rosa-Ferreira and Munro, Dev Cell, 2011) to enhance recruitment to lysosomes.

While this point is well taken, however, we believe that the added siRNA depletion of kinesin-1 demonstrate the specificity of QPD-OTf for kinesin-1 and feel that redirecting kinesin-activity away from the Golgi would be redundant with the brefeldin treatment that shows that Golgi fragmentation leads to an augmentation in nucleation sites.

Reviewer #3 (Remarks to the Author):

In the manuscript entitled "Kinesin-1 motility traced by an activity-based precipitating dye" by Angerani et al., the authors reported the discovery of a fluorogenic substrate (QPD-OTf) for tracing the activity of kinesin-1 in vivo. Importantly, the authors showed that the substrate fluorogenic substrate acts as an ATP analogue and binds to Kinesin-1 to yield an insoluble fluorescent dye that decorates on the path traveled by Kinesin-1. Overall, this is a very interesting work and potentially significant for the kinesin field. Before this manuscript can be considered for publication, the following comments need to be sufficiently addressed.

1. The Abstract in the present form does not concisely summarize the major findings of the work and the implication(s) of these findings. The authors should consider rewriting a new Abstract by combining Lines 14-18 (of the current Abstract) and the majority of the last paragraph of the Introduction (Lines 51-60).

The abstract has been modified to better summarize the major findings and implications of the work.

2. Captions of many figures need to be revised to make them more informative, which will help the readers better appreciate the results. For some figures, the panels need to be reordered too.

As raised by reviewer 2 (point 6), captions have now been changed to include sample size, number of replicates, statistical tests. The panels of figures have also been updated with the additional data.

For Figure 1, change the title to "Schema of the soluble profluorophore QPD-OTf and the insoluble fluorescent dye QPD", and remove the panel on the right.

The title has now been changed and the figure has been updated.

For Figure 2, change the caption to "QPD-OTf forms crystals that strongly colocalize with MTs in living cells."

The caption of Fig.2 has been modified according to the suggestion.

For Figure 3, change the title to "Formation of QPD crystals in U2OS live cells is disrupted by induced microtubule stabilization or depolymerization". For Fig. 3A, the authors should consider writing "Representative images of crystal formation in cells treated (left) with 1 μ M Taxol for 1 hour and 20 μ M QPD-OTf for 4 hours at 37 C; (middle) on ice for 1 hour and with 20 μ M QPD-OTf for 4 hours on ice; and (right) with 20 μ M QPD-OTf for 4 hours at 37 C. (Bottom) Zoomed-in images of cells in the black squares". To better help the readers comprehend Fig. 2A, the authors should use dashed lines to link the original boxes and the corresponding zoomed-in images.

The title of Fig.3 has been changed, the caption modified, and dashed lines to link boxes to the corresponding zoomed-in images added.

For Figure 4, change the title to "The nucleation center of the QPD crystals is localized at the Golgi apparatus". Similar to Fig. 3A, the text for Fig. 4A should be revised to make it more descriptive and informative for the readers.

The title of Fig.4 has been changed. The figure legend has been updated to be more descriptive.

For Figure 5, change the title to “Kinesin-1 activity is required for QPD crystal formation in vivo”. The authors should also add the control images for Figs. 5A and B, and add dashed lines to link the white boxes and the corresponding zoomed-in images in Fig. 5B.

The title of Fig.5 has been changed; dashed lines to link the zoomed-in boxes to the corresponding image have been added. The control image is now included in Fig.S9

For Figure 6, I suggest the authors to swap Fig. 6A and Fig. 6C so that the table of conditions for the 6 in vitro precipitation experiments of QPD is presented first, followed by the images of the 6 samples and the intensity readouts of all 6 samples.

We thank the reviewer for this comment, Panels A and C of Fig. 6 have been swapped as suggested.

For Supplementary Figure 6, change the title to “The centrosome does not exhibit colocalization with the nucleating site for QPD precipitation”.

The title of Fig.S6 (now Fig.S10) has been changed.

3. In Lines 96-99, the authors need to consider revising the text to “We first treated U2OS cells with 1 μ M Taxol and 20 μ M QPD-OTf for 4 hours and found that compared to the control experiments (Fig. 3A, right), Taxol-induced MT stabilization reduced the crystal formation by 75% (Fig. 3A, left; and Fig. 3B). I suggest the authors to re-order the image sequence to “Control”, “Taxol” and “On ice” (from left to right).

The text of has been revised. NB line number have now changed.

4. In Line 101, please clearly indicate that the middle panel of Fig. 3A is the image that corresponds to the statement “no crystals were observed (Fig. 3)”.

The text has been corrected as suggested. NB line number have now changed.

5. In Line 115, please clarify the operation of sequential treatment of QPD-OTf. Did the authors mean “subsequent” treatment of QPD-DTf?

The text has been corrected as suggested.

6. In Fig. 5C, the authors referred to the two kinesin constructs as Kin560 and Kin330, but in all other places of the manuscript, these two constructed were referred to kin330 (Lines 141, 159 and 466) and kin560 (Lines 141, 146-148 and 467). To be consistent and avoid confusion, the authors need to use the same naming convention for both constructs throughout the manuscript.

We thank the reviewer for point out this discrepancy. If should have read Kin560 and Kin330 throughout the manuscript. This has now been corrected.

7. In Lines 171-173, Fig. 8A-C should be Fig. 6A-C.

The text has been corrected. NB line number have now changed.

8. In Line 179, change the subsection heading to “QPD-OTf is a substrate analogue of ATP.”

The subsection heading has been changed.

9. In the Section entitled “QPD-OTf conversion to QPD depends on kinesin-1 motility” (Lines 137-161), several important observations were made that are key to understand the role of kinesin-1 activity in the conversion of QPD-OTf to QPD. The authors need to expand this section to explain: 1) why transfection with the immotile Kin330 reduced the number of crystals by 87% compared with the non-transfected control cells; and 2) why activation of kinesin-1 with kinesore does not mimick transfection with Kin560 in terms of crystal formation.

1) Kin330 is immotile and does not turnover ATP. The text was modified for clarity: "In cells transfected with the immotile Kin330 the number of crystals was reduced by 87% compared to non-transfected cells (Fig. 5A, Supplementary Fig. 14, and Fig. 5C), consistent with the immotile function of Kin330"

2) While the addition of kinesore functionally mimicks kin560 (cargo-independent motor activity), the experiments are different. Transfection of Kin560, increases the number of kinesin-1 in a cell and we have the is residual activity of endogenous kinesin-1 that is free to produce crsytals as in untreated cells. In the case of kinesore addition, the endogenous kinesin-1 is monopolized by kinesore, hence only the diffuse formation of precipitate outside of the Golgi-associated MT is observed.

10. The Discussion needs to be significantly improved to help the readers better appreciate the results and the underlying mechanisms and the significance and implication(s) of this work. I suggest that authors to: 1) move up the last paragraph of the Discussion to become the starting paragraph; 2) add one or two paragraphs to discuss about the molecular mechanisms of some key observations that may not be immediately clear to the readers; and 3) add at least one more paragraph to discuss how others can take advantage of the fluorogenic substrate in studying kinesin-1 activity in vivo and how similar fluorogenic substrates specific for other kinesin motors such as kinesin-5 can be rationally designed.

We very much appreciate the comments. 1) We had organized the section to put the work in the context of other fluorogenic probes and highlight the fact that probes that can detect motor activity and trace the motion have never been reported. We understand the opinion of the reviewer but we think it is appropriate to keep it in this order. 2) We believe that it remains premature to speculate on the molecular mechanism given the lack of a co-crystal structure of QPD-OTf with the motor domain of kinesin-1. 3) A paragraph highlights the opportunities arising from tracking kinesin-1 with a small molecule rather than with genetic constructs or immunostaining. The work highlights that is possible to track a motor protein with a precipitating dye which should certainly inspire the search for a substrate for kinesin-5. Whether this will come from rational design or screening of potential substrates or a combination is pure speculation. Given the journal's instruction for a short discussion, we think this speculation extends beyond the aim of the discussion.

11. In Lines 219-220, the authors stated that "kinesin-1 inhibitors disrupt the formation of the crystals." Are the authors referring to published work by others or results in this manuscript?

This is another excellent point. The sentence was intended to refer to the experiment with Kin330 however, we agree that the use of "inhibitor" was misleading. Based on the additional experiment knocking down kinesin-1, we have changed the sentence to "kinesin-1 depletion disrupt the formation of crystals."

Reviewers' Comments:

Reviewer #1:

Remarks to the Author:

I think the authors have answered most of my questions well. However, for reviewer 1, question 7, the author said that different cell lines had different crystal morphology and this answer is not very related to the question. As in figures 2&4&5, the authors used the same cell U2OS and showed different crystal morphology. Maybe the authors meant that even different cells have different crystal morphology. If so, please provide possible explanation.

Reviewer #2:

Remarks to the Author:

The requirement for kinesin-1 in crystal formation in cells is better demonstrated. The improved imaging in Figure 2D makes a better case that the fluorescence signal is, at least in part, microtubule associated.

However, there are still a big limitation in how far this work can be interpreted with respect to 'tracing' kinesin motility. Case in point is figure 1C - the crystals apparently penetrate the nucleus. Microtubules don't penetrate through the nucleus. What aspect of kinesin-1-microtubule motility can this possibly be tracing? Is the more likely explanation that the crystals simply grow as the substrate concentration increases. Perhaps they grow from the Golgi region because there is a lot of kinesin activity there, but there is insufficient evidence that the paths the crystals take are kinesin-1 tracks.

I would urge the authors to use caution in making this 'tracing' claim.

Minor point:

Label on panel 5F should be Kif5B.

Reviewer #3:

Remarks to the Author:

In this revision, the authors have sufficiently addressed all my original comments with the exception of comment #9. In their response to the first part of comment #9, the authors suggest transfection with the Kin330 reduces the number of crystals by 87% compared with the non-transfected control cells because Kin330 is immotile and does not hydrolyze ATP. Based on the published work by Schepis et al (Cellular Microbiology, 2007 9: 1960-1973), I am not convinced that Kin330 — a truncated kinesin-1 construct that contains the first 330 amino acids — necessarily lacks the ability to hydrolyze ATP. The mechanism underlying the reduction of crystal formation due to Kin330 transfection is likely very complex; one possible scenario is that Kin330 inhibits the endogenous kinesin-1 to reduce crystal formation (Schepis et al, Cellular Microbiology, 2007 9: 1960-1973). If the authors can confirm that Kin330 inhibits kinesin-1 motility, the transfection experiments with Kin330 are sufficient to show that QPD-OTf conversion to QDP depends on kinesin-1 motility. In the section entitled "Kinesin-1 forms QPD crystals in vitro", the authors should also include experiments of crystal formation in the presence of both kinesin-1, microtubules and Kin330, as these experiments will likely reveal the mechanism underlying the effects of Kin330 transfection on crystal formation in vivo. The transfection experiments with Kin560 are compounded by lack of a definitive understanding of how Kin560 affects kinesin-1 motility in vivo, and the authors should consider not including these experiments.

POINT BY POINT RESPONSE TO THE REVIEWER'S COMMENTS

Reviewer #1 (Remarks to the Author):

I think the authors have answered most of my questions well. However, for reviewer 1, question 7, the author said that different cell lines had different crystal morphology and this answer is not very related to the question. As in figures 2&4&5, the authors used the same cell U2OS and showed different crystal morphology. Maybe the authors meant that even different cells have different crystal morphology. If so, please provide possible explanation.

Indeed, different cell lines give slightly different crystal pattern. Within the same cell line, we generally observe the same pattern of crystal formation. The appearance of this pattern will depend on the time of crystal formation and imaging method. U2OS are used exclusively in Figure 2A (Figure 2B-F use PTK2 and HeLa). While figure 4 and 5 use exclusively U2OS, comparison between Fig 2A and Fig 4-5 should be treated with some caution because Fig 2A was acquired after a MeOH fixation (for immunostaining) which will partially dissolve crystals, hence it is expected to observe a less extensive pattern of crystal formation. Fig 2A was confusing because the nucleus was stained with DAPI and both the crystals and DAPI are imaged in the same channel. For clarity, a yellow contour line has now been added around the nucleus to highlight the fact that this fluorescence is not from crystals.

Reviewer #2 (Remarks to the Author):

The requirement for kinesin-1 in crystal formation in cells is better demonstrated. The improved imaging in Figure 2D makes a better case that the fluorescence signal is, at least in part, microtubule associated.

We thank the referee for noting the improvements to the manuscript.

However, there are still a big limitation in how far this work can be interpreted with respect to 'tracing' kinesin motility. Case in point is figure 1C - the crystals apparently penetrate the nucleus. Microtubules don't penetrate through the nucleus. What aspect of kinesin-1-microtubule motility can this possibly be tracing? Is the more likely explanation that the crystals simply grow as the substrate concentration increases. Perhaps they grow from the Golgi region because there is a lot of kinesin activity there, but there is insufficient evidence that the paths the crystals take are kinesin-1 tracks.

We agree with the reviewer that MTs do not penetrate through the nucleus and the observation of crystals going through the nucleus is an artefact of sample preparation for the FIB imaging. For imaging crystals with FIB-SEM, we used the cell line that produce most crystals (HeLa) and with a prolonged 4h QPD-OTf incubation time to ensure maximum crystal display in cells. HeLa cells are then fixed and dehydrate which undoubtedly lead to some motion in the organelles. With our live imaging we noted that existing crystals deform the plasma membrane when cells migrate, indicating the rigidity of the crystals Fig 1B (HeLa). A similar observation can be made in Fig 2A where the end of the crystal is deforming the membrane. For clarity, the following sentence has been added in the main text: "It should be

noted that the rigidity of crystals is such that plasma membranes of retracting cells are deformed (Fig. 1B). In the FIB-SEM image, a more extreme case is observed where the crystal penetrates through the nucleus (Fig. 1C and Supplementary Fig 6A). Given the incubation time, fixation, and dehydration steps involved in the sample preparation, this observation may be an artefact of sample preparation”

The claims with respect to ‘tracing’ come from the fact that the same QPD precipitating dye unmasked from different precursors (such as the ELF-97, a commercial product) do not give any crystals in the form of fibres as we observed in the present work. Five different precursors of this QPD dye have been reported (refs. 22, 23, 25, 26, 28 in the main text), and the QPD dye was revealed using phosphatase, proteases, H₂O₂ or ruthenium catalysed reactions. In none of the cases the precipitation event led to the generation of crystal fibres. Thus, our observed fibre formation cannot be an intrinsic product of the QPD itself. In addition, as shown by the FIB-SEM images, the crystals show a clear helicity, a characteristic that cannot simply arise from an achiral molecule (QPD). Thus, the unique image obtained with QPD-OTf are clearly a product of the enzyme that yield QPD rather than a product of QPD itself. We clearly demonstrate that kinesin-1 is necessary and sufficient to convert QPD-OTf to QPD. It stands to reason that the fibre is a product of kinesin-1 motility, effectively tracing kinesin-1 motion. Orthogonal evidence of this also comes from images of Kin560-GFP where a strong co-localization is observed between the tubulin-bound Kin-560 and the crystals, and increased kinesin-1 activity increases crystal formation (Fig. 5C).

Minor point:

Label on panel 5F should be Kif5B.

The label on panel 5F has now been corrected.

Reviewer #3 (Remarks to the Author):

In this revision, the authors have sufficiently addressed all my original comments with the exception of comment #9. In their response to the first part of comment #9, the authors suggest transfection with the Kin330 reduces the number of crystals by 87% compared with the non-transfected control cells because Kin330 is immotile and does not hydrolyze ATP. Based on the published work by Schepis et al (Cellular Microbiology, 2007 9: 1960-1973), I am not convinced that Kin330 — a truncated kinesin-1 construct that contains the first 330 amino acids — necessarily lacks the ability to hydrolyze ATP. The mechanism underlying the reduction of crystal formation due to Kin330 transfection is likely very complex; one possible scenario is that Kin330 inhibits the endogenous kinesin-1 to reduce crystal formation (Schepis et al, Cellular Microbiology, 2007 9: 1960-1973). If the authors can confirm that Kin330 inhibits kinesin-1 motility, the transfection experiments with Kin330 are sufficient to show that QPD-OTf conversion to QPD depends on kinesin-1 motility. In the section entitled “Kinesin-1 forms QPD crystals in vitro”, the authors should also include

experiments of crystal formation in the presence of both kinesin-1, microtubules and Kin330, as these experiments will likely reveal the mechanism underlying the effects of Kin330 transfection on crystal formation in vivo. The transfection experiments with Kin560 are compounded by lack of a definitive understanding of how Kin560 affects kinesin-1 motility in vivo, and the authors should consider not including these experiments. We thank the referee for the considerations regarding our revision. The mechanism behind the effects of Kin330 transfection on crystal formation in cells is certainly a very good point and we agree with the reviewer that this could be an interesting area to investigate, however, we believe these studies fall beyond the scope of this paper.

As reported by the work of Schepis et al. (*Cellular Microbiology*, 2007 9: 1960-1973), Kin330-GFP overexpression is known to “functionally inhibit kinesin-1”. While we recognize that, as state by Schepis et al., “The way kin330-GFP inhibited this motor complex was less clear. [...] A possibility is that the construct binds to the C-terminal tail of KHC”, the reduction in number of crystals observed in the sample overexpressing Kin330-GFP is in line with the reported functional inhibition of kinesin-1.

In order to avoid ambiguity, the main text has been changed and the text now reads: “Cells transfected with the kinesin-1 mutant Kin330 showed a reduction in crystal numbers by 87% compared to non-transfected cells (Fig. 5A, Supplementary Fig. 15, and Fig. 5C), consistent with the inhibitory effect of Kin330 on the functional activity of native kinesin-1”.

Regarding Kin560, we thank the reviewer for the suggestion, however we wish to keep the experiment. The experimental outcome with the established Kin560 construct (P.J. Hooikaas et al., *JCB*, 2019; R.B. Case, et al. *Cell*, 1997; A. Padzik et al. *Front. Cell. Neurosci.* 2016) represents the counterpart to the Kin330 case and in the absence of a better mutant with a well-defined molecular mechanism, we think this experiment should be presented. Furthermore, Kin560-GFP yield a strong co-localization with the crystal, lending further evidence to the link between crystal and kinesin.